# An ancient germ cell-specific RNA-binding protein protects the germline from cryptic splice site poisoning

Ingrid Ehrmann[1], James H Crichton[2], Matthew R Gazzara[3,4], Katherine James[5], Yilei Liu[1,6], Sushma Nagaraja Grellscheid[1,7], Tomaž Curk[8], Dirk de Rooij[9,10], Jannetta S Steyn[11], Simon Cockell[11], Ian R Adams[2], Yoseph Barash[3,12]*, David J Elliott[1]*

[1]Institute of Genetic Medicine, Newcastle University, Newcastle, United Kingdom; [2]MRC Human Genetics Unit, MRC Institute of Genetics and Molecular Medicine, University of Edinburgh, Edinburgh, United Kingdom; [3]Department of Genetics, Perelman School of Medicine, University of Pennsylvania, Philadelphia, United States; [4]Department of Biochemistry and Biophysics, Perelman School of Medicine, University of Pennsylvania, Philadelphia, United States; [5]Life Sciences, Natural History Museum, London, United Kingdom; [6]Department of Plant and Microbial Biology, University of Zürich, Zürich, Switzerland; [7]School of Biological and Biomedical Sciences, University of Durham, Durham, United Kingdom; [8]Laboratory of Bioinformatics, Faculty of Computer and Information Sciences, University of Ljubljana, Ljubljana, Slovenia; [9]Reproductive Biology Group, Division of Developmental Biology, Department of Biology, Faculty of Science, Utrecht University, Utrecht, The Netherlands; [10]Center for Reproductive Medicine, Academic Medical Center, University of Amsterdam, Amsterdam, The Netherlands; [11]Bioinformatics Support Unit, Faculty of Medical Sciences, Newcastle University, Newcastle, United Kingdom; [12]Department of Computer and Information Science, University of Pennsylvania, Philadelphia, United States

*For correspondence:
yosephb@seas.upenn.edu (YB);
David.Elliott@ncl.ac.uk (DJE)

**Competing interests:** The authors declare that no competing interests exist.

**Abstract** Male germ cells of all placental mammals express an ancient nuclear RNA binding protein of unknown function called RBMXL2. Here we find that deletion of the retrogene encoding RBMXL2 blocks spermatogenesis. Transcriptome analyses of age-matched deletion mice show that RBMXL2 controls splicing patterns during meiosis. In particular, RBMXL2 represses the selection of aberrant splice sites and the insertion of cryptic and premature terminal exons. Our data suggest a *Rbmxl2* retrogene has been conserved across mammals as part of a splicing control mechanism that is fundamentally important to germ cell biology. We propose that this mechanism is essential to meiosis because it buffers the high ambient concentrations of splicing activators, thereby preventing poisoning of key transcripts and disruption to gene expression by aberrant splice site selection.
DOI: https://doi.org/10.7554/eLife.39304.001

## Introduction

It has been a long recognised hallmark of mammalian gene expression patterns that there are extremely high levels of transcription and transcriptome complexity in testicular cells (*de la Grange et al., 2010*; *Licatalosi, 2016*; *Soumillon et al., 2013*; *Pan et al., 2008*; *Wang et al., 2008*; *Clark et al., 2007*; *Grosso et al., 2008*; *Yeo et al., 2004*). These high gene expression levels are

**eLife digest** In humans and other mammals, a sperm from a male fuses with an egg cell from a female to produce an embryo that may ultimately grow into a new individual. Sperm and egg cells are made when certain cells in the body divide in a process called meiosis. Many proteins are required for meiosis to happen and these proteins are made using instructions provided by genes, which are made of a molecule called DNA.

The DNA within a gene is transcribed to make molecules of ribonucleic acid (or RNA for short). The cell then modifies many of these RNAs in a process called splicing before using them as templates to make proteins. During splicing, segments of RNA known as introns are discarded and other segments termed exons are joined together. Some exons may also be removed from RNAs in different combinations to create different proteins from the same gene.

A protein called RBMXL2 is able to bind to RNA molecules and is only made during and after meiosis in humans and most other mammals. RBMXL2 can also bind to other proteins that are known to be involved in controlling splicing of RNAs, but its role in splicing remains unclear.

To address this question, Ehrmann et al. studied the gene that encodes the RBMXL2 protein in mice. Removing this gene prevented male mice from being able to make sperm. Further experiments using a technique called RNA sequencing showed that the RBMXL2 protein helps to ensure that splicing happens correctly by preventing bits of exons and introns in mouse genes from being rearranged. These findings suggest that the gene encoding RBMXL2 is part of a splicing control mechanism that is important for making sperm and egg cells.

The work of Ehrmann et al. could eventually help some couples understand why they have problems conceiving children. Male infertility is poorly understood, and not knowing its causes can harm the mental health of affected men. Furthermore, these findings may help researchers to understand the role of a closely related protein called RBMY that has also been linked to infertility in men, but is much more difficult to study.

DOI: https://doi.org/10.7554/eLife.39304.002

thought to result from epigenetic changes that favour relaxed patterns of gene expression during meiosis – the unique form of division used to generate sperm and eggs (*Soumillon et al., 2013*). Many nuclear RNA-binding proteins are differentially expressed during and immediately after meiosis (*Grellscheid et al., 2011*; *Schmid et al., 2013*). These include the nuclear RNA binding protein RBMXL2 (also known as hnRNP GT) that is expressed only during and immediately after male meiosis, but not in the preceding spermatogonial cells (*Ehrmann et al., 2008*) (*Figure 1A*). Consistent with an important function in germ cell development, genetic studies have identified point mutations within infertile men in the human chromosome 11 *RBMXL2* gene (*Westerveld et al., 2004*). A further connection to male infertility is that *RBMXL2* belongs to the same gene family as *RBMY*, which was historically the first human Y chromosome infertility gene identified in the search for the *AZF* (*AZOO-SPERMIA FACTOR*) gene (*Ma et al., 1993*).

*RBMXL2* evolved ~65 million years ago via retrotransposition of an mRNA from the X chromosome located RBMX gene (*Figure 1A*). An *RBMXL2* gene is found in all placental mammals, which is consistent with a fundamental role in germ cell biology. This role remains to be identified, but RBMXL2 protein has an N-terminal RNA Recognition Motif (abbreviated RRM, *Figure 1B*). RBMX and RBMY proteins are also nuclear RNA binding proteins that are very similar to RBMXL2 (73.2% and 36.8% overall identity to RBMX and RBMY, respectively; 93.7% and 77.2% identity within the RRMs, *Figure 1—figure supplement 1*). The RNA binding specificity of RBMXL2 protein is unknown, but both RBMX and RBMY proteins bind to AA dinucleotide-containing RNA sequences (*Cléry et al., 2011*; *Moursy et al., 2014*; *Nasim et al., 2003*). The RBMXL2, RBMX and RBMY proteins interact with and modulate the splicing activity of Tra2β and SR proteins in vitro (*Figure 1B*) (*Cléry et al., 2011*; *Moursy et al., 2014*; *Liu et al., 2009*; *Nasim et al., 2003*; *Elliott et al., 2000a*), suggesting a role in splicing control. Maintaining proper ratios of mRNA splice isoforms can be critical in normal development (*Kalsotra and Cooper, 2011*), where changes in isoforms can have effects on encoded proteins ranging from major to subtle. Alternative splicing is known to be critical for germ cell development. For example, deletion of the splicing regulator protein PTBP2 within

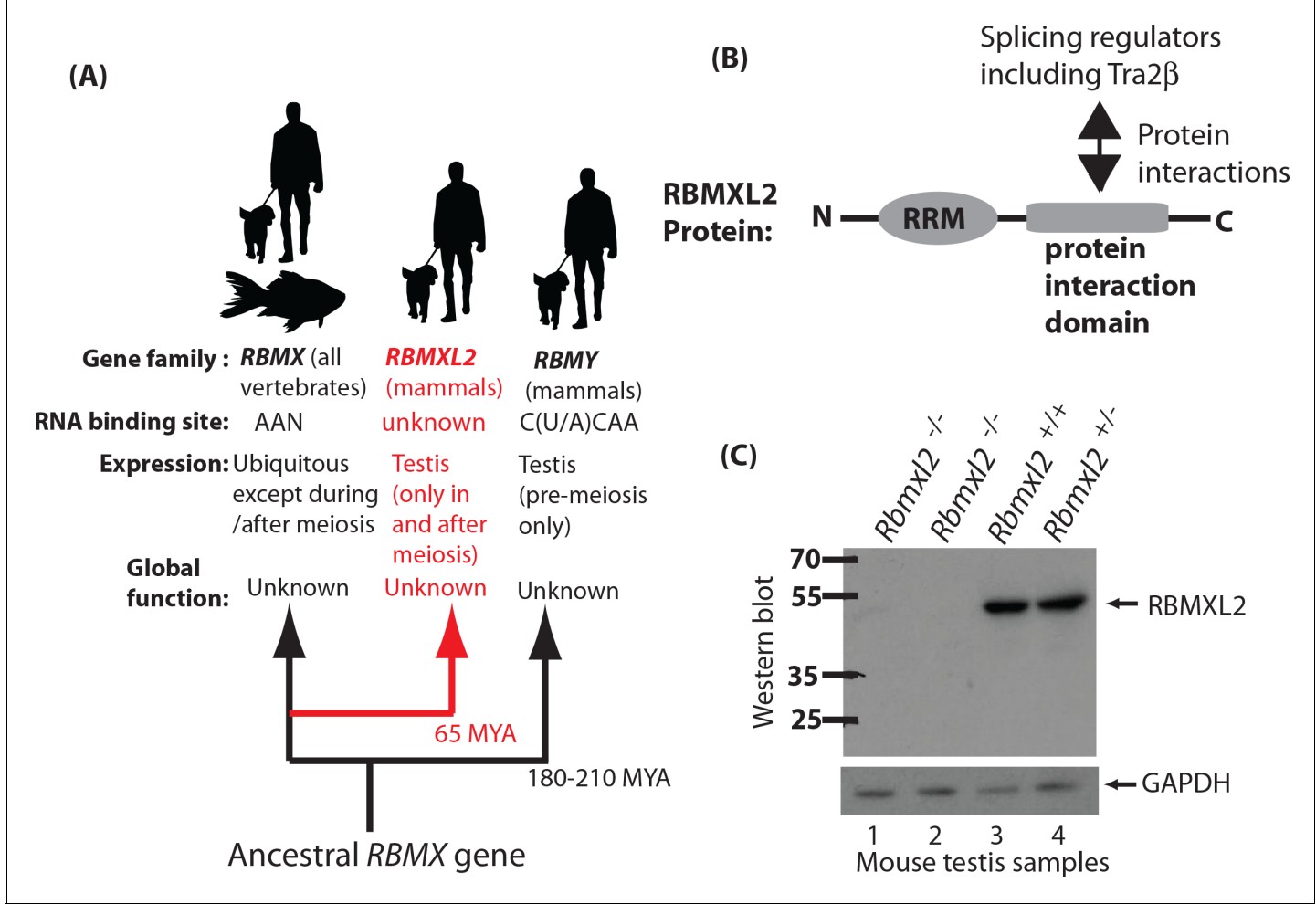

**Figure 1.** Creation of a mouse model which does not express the testis-specific RNA binding protein RBMXL2. (**A**) The *RBMXL2* gene evolved via retrotransposition of *RBMX* early in mammalian evolution. The cladogram shows three members of the gene family that derived from an ancestral *RBMX* gene, summarising their global functions, expression, and RNA binding sites of their encoded proteins. (**B**) Modular structure of the RBMXL2 protein, showing the known protein interaction with Tra2β. (**C**) Western blot confirming no RBMXL2 protein is made within 2 replicate samples of *Rbmxl2*[-/-] adult testis, compared to wild type (*Rbmxl2*[+/+]) and heterozygous (*Rbmxl2*[+/-]) adult testes.

DOI: https://doi.org/10.7554/eLife.39304.003

The following figure supplements are available for figure 1:

**Figure supplement 1.** Alignment of RBMXL2 with RBMX and RBMY.

DOI: https://doi.org/10.7554/eLife.39304.004

**Figure supplement 2.** Genomic organisation of different genotype mice.

DOI: https://doi.org/10.7554/eLife.39304.005

germ cells affects mRNA isoforms important for cell-cell communication with Sertoli cells (*Hannigan et al., 2017*). Some alternative splicing events in the testis are conserved between humans and mice so may control fundamental aspects of germ cell biology (*Schmid et al., 2013*). However many alternative splicing patterns are not conserved between humans and mice (*Kan et al., 2005*). RBMX protein is also reported to control transcription (*Takemoto et al., 2007*), affect DNA double strand repair and mitotic sister chromatid cohesion (*Adamson et al., 2012*; *Matsunaga et al., 2012*), and to bind to m6A methylated RNA (*Liu et al., 2017*).

The function of RBMXL2 and why this RNA binding protein has been conserved across placental mammals are not known. A major factor limiting understanding of endogenous RBMXL2 functions has been the absence of a reliable mouse model. Development of a mouse model is also critical to

test the importance of this wider family of RNA binding proteins in germ cell development. Men carrying the *AZFb* deletion on their Y chromosomes are missing *RBMY* genes and undergo meiotic arrest. However, it is unclear if *RBMY* loss is causing this phenotype, because the deletion interval encompasses several other genes which could contribute to male infertility (*Elliott, 2000*; *Vogt et al., 1996*). Within the meiotic and immediately post-meiotic germ cells which express RBMXL2, the *RBMX* and *RBMY* genes are transcriptionally inactivated within a heterochromatic structure called the XY body (*Wang, 2004*). Meiosis thus provides a genetically tractable window to probe RBMXL2 function where there should be no redundancy effects possible with either RBMX or RBMY. Hence to discover what RBMXL2 does in the germline we have made a conditional *Rbmxl2* gene knockout mouse. Analysis of this knockout mouse reveals that RBMXL2 protein is essential for meiosis and has a major role in protecting the meiotic transcriptome from aberrant selection of cryptic splice sites that are normally ignored by the spliceosome. Our data suggest this fundamentally important process operates so efficiently in meiosis that it has been previously undetected, yet is critical to avoid male infertility caused by aberrant splicing of key meiotic transcripts.

## Results

### RBMXL2 protein is essential for male fertility

To test how important RBMXL2 protein is for male germline development we made a conditional mouse model in which we flanked the *Rbmxl2* open reading frame with *LoxP* sites. Since *Rbmxl2* is only expressed in the testis (*Elliott et al., 2000b*), we chose to delete the entire *Rbmxl2* open reading frame to create a null allele. We achieved this by crossing our conditional model with a mouse strain expressing *Cre* recombinase under control of the ubiquitous *Pgk* promoter (experimental details are provided in the Materials and methods). We confirmed deletion of this genomic region in homozygous *Rbmxl2* gene knockout (*Rbmxl2*[-/-]) mice by Southern blotting (*Figure 1—figure supplement 2*) and the specific absence of the 50 KDa RBMXL2 protein from knockout testes by Western blotting (*Figure 1C*).

*Rbmxl2*[-/-] mice developed comparably to their wild type littermates but their testes were much smaller (*Figure 2A and B*). This small testis phenotype correlated with a severe disruption of testicular histology. Adult *Rbmxl2*[-/-] mice contained cells undergoing meiosis but almost no post-meiotic cells (*Figure 2C*, meiotic spermatocytes are abbreviated Spc, and post-meiotic round spermatids are abbreviated Rtd). The epididymis dissected from *Rbmxl2*[-/-] mice were completely devoid of sperm. Four wild type mice tested had an average of $7.07 \pm 1.39 \times 10^6$ epididymal sperm ml$^{-1}$, compared with zero in four *Rbmxl2*[-/-] mice (*Figure 2D and E*). No effect on female fertility was observed in *Rbmxl2*[-/-] mice (not shown).

Very rarely a few round spermatids were observed in adult *Rbmxl2*[-/-] testis sections (*Figure 2—figure supplement 1A*), although no elongated spermatids were detected. The presence of these round spermatids indicates that meiosis can occasionally complete in the absence of RBMXL2 protein. This is consistent with loss of RBMXL2 causing either (1) a developmental block in meiosis from which a few cells can escape; or alternatively (2) a slow attrition effect, in which the *Rbmxl2*[-/-] testis phenotype is caused by post-meiotic stages dying in the adult testis. To differentiate between these two possibilities we histologically analysed testes at 21 days postpartum (21dpp), at which point germ cells in the wild type (*Rbmxl2*[+/+]) mice have just started to enter the post-meiotic round spermatid stage. At 21dpp there were still significantly fewer round spermatids in the *Rbmxl2*[-/-] sections compared to wild type (*Figure 2—figure supplement 1B*), even though there would have been less time for round spermatid cell death compared with the adult. This result thus supports a strong meiotic block in *Rbmxl2*[-/-] mice, with occasional completion of meiosis rather than a gradual attrition of round spermatids.

### Most *Rbmxl2*[-/-] germ cells do not progress past meiotic diplotene

The above data demonstrate that an *Rbmxl2* gene, which is conserved in all placental mammals and specifically expressed in male meiosis, is essential for mouse spermatogenesis somewhere within meiotic prophase. Staining of nuclear spreads with antibodies specific to the meiotic chromosome proteins SYCP1 and SYCP3 more precisely showed that *Rbmxl2*[-/-] mice arrest germ cell development during the diplotene substage of meiotic prophase (*Figure 2F* and *Figure 2—figure supplement 2*).

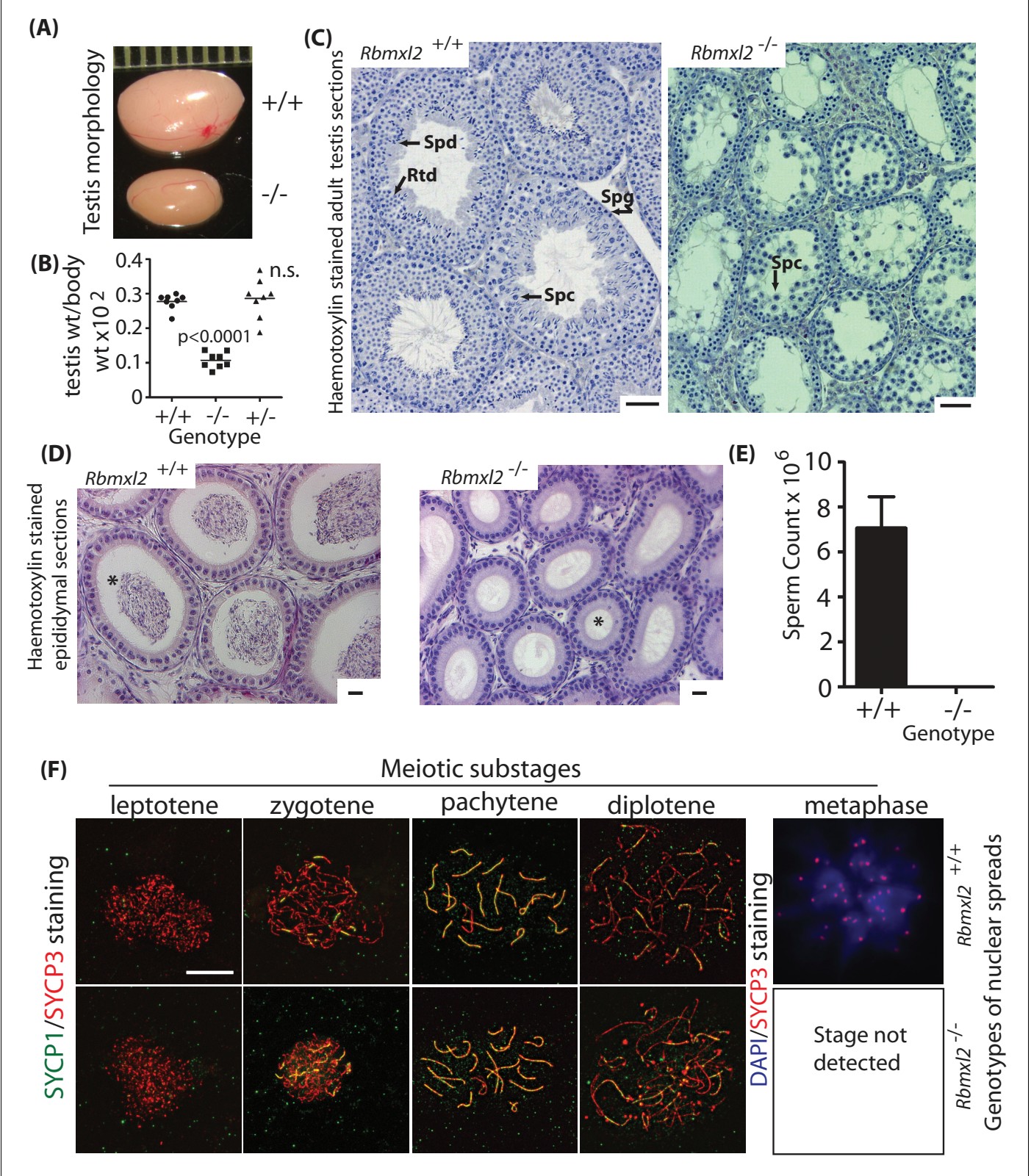

**Figure 2.** RBMXL2 protein is important for mouse germ cells to progress past diplotene into metaphase I of meiosis. (**A**) Testis morphologies of adult wild type and homozygous knockout (*Rbmxl2⁻/⁻*) mice (scale above testes shown in millimeters). (**B**) Testis:body weight ratios of different genotype mice. P values were calculated using a t test. (**C**) Micrographs of haemotoxylin-stained testis sections from wild type and *Rbmxl2⁻/⁻* mice. Abbreviations: Spg, spermatogonia; Spc, spermatocyte; Rtd, round spermatid; Spd, elongating spermatid. Scale bar = 20 um. (**D**) Epididymis of wild type and *Rbmxl2⁻/⁻*

*Figure 2 continued on next page*

*Figure 2 continued*

mice stained with haemotoxylin and eosin. Asterisk indicates lumen (note lumen is empty in *Rbmxl2*<sup>-/-</sup> section indicative of azoospermia, but contains multiple cells in the wild type). Scale bar = 20 um. (**E**) Epididymal sperm counts of wild type and *Rbmxl2*<sup>-/-</sup> mice (n = 4 of each genotype). The error bar represents SEM. (**F**) Meiotic prophase I stages detected in wild type and *Rbmxl2*<sup>-/-</sup> testis chromosome spreads stained for SYCP3 (pseudocoloured red) and SYCP1 (pseudocoloured green); and DAPI and SYCP3 (pseudocoloured blue and red respectively). No metaphase I nuclei were identified in the *Rbmxl2*<sup>-/-</sup> testes. Quantitative data are provided in *Figure 2—figure supplement 2*.

DOI: https://doi.org/10.7554/eLife.39304.006

The following figure supplements are available for figure 2:

**Figure supplement 1.** Germ cells occasionally progress through meiosis into the round spermatid stage.
DOI: https://doi.org/10.7554/eLife.39304.007
**Figure supplement 2.** Graphical analysis of meiotic chromosome spreads.
DOI: https://doi.org/10.7554/eLife.39304.008
**Figure supplement 3.** Detailed histological analysis of *Rbmxl2*<sup>-/-</sup> testes.
DOI: https://doi.org/10.7554/eLife.39304.009
**Figure supplement 4.** Organisation of H3K9me3 staining and centromere clusters within wild type and *Rbmxl2*<sup>-/-</sup> mice show defects arise during pachytene but become progressively worse by diplotene.
DOI: https://doi.org/10.7554/eLife.39304.010

Metaphase I nuclei were only detected in wild type mice and not in *Rbmxl2*<sup>-/-</sup> mice (N = 3 wild type and N = 3 *Rbmxl2*<sup>-/-</sup>testes, 180 and 138 spermatocyte nuclei scored for wild type and *Rbmxl2*<sup>-/-</sup> respectively). No significant differences in the frequency of earlier stages of meiotic prophase were detected between *Rbmxl2*<sup>-/-</sup> and wild type testes. The presence of germ cell populations up to diplotene in *Rbmxl2*<sup>-/-</sup> testes was further confirmed by analysis of histological sections stained with Periodic Acid Schiff (PAS) (*Figure 2—figure supplement 3A–D*). Staining of adult testis sections with PAS or antibodies specific to the apoptotic marker activated Caspase three further showed that adult *Rbmxl2*<sup>-/-</sup> germ cells die via apoptosis (*Figure 2—figure supplement 3E–F*).

Analysis of diplotene nuclear spreads revealed further abnormalities in the *Rbmxl2*<sup>-/-</sup> testis. There was a decreased number of H3K9me3-marked centromere clusters (*Takada et al., 2011*), with each individual cluster also containing more centromeres than in wild type testis (*Figure 2—figure supplement 4A–C*). H3K9me3 staining was also present at the sex body in a proportion (82%) of wild type diplotene spermatocytes, but essentially absent in mutant diplotene spermatocytes (1.5% of diplotene spermatocytes stained, 66 *Rbmxl2*<sup>-/-</sup> and 66 wild type nuclei scored, n = 3 for both) (*Figure 2—figure supplement 4D*). These defects in centromere clustering and H3K9me3 modification of the sex body were still detectable but less severe at pachytene (*Figure 2—figure supplement 4D*). Asynapsis was rarely observed in either control or mutant pachytene stage spermatocytes, indicating that the defect causing diplotene arrest does not significantly impact either synaptonemal complex formation or meiotic homology searching (*Figure 2—figure supplement 4E*) (N = 3, 109 and 94 nuclei scored for wild type and *Rbmxl2*<sup>-/-</sup> respectively).

In summary, the above mouse phenotype showed that deletion of the ancient RNA binding protein RBMXL2 induces progressive defects in male mouse meiotic prophase. These defects culminate with a major block during diplotene that prevents entry into metaphase I, but with defects becoming already apparent during pachytene.

## Gene expression analysis of age matched wild type and *Rbmxl2*<sup>-/-</sup> testes

Since RBMXL2 is a nuclear RNA binding protein we predicted that the above phenotype could be associated with a primary molecular defect in generating or processing RNAs in the *Rbmxl2*<sup>-/-</sup> testis. To test this we analysed wild type and *Rbmxl2*<sup>-/-</sup> testes using RNAseq. Based on the knockout phenotype above, and because the adult wild type testis contains additional more advanced germ cells that are missing from the adult *Rbmxl2*<sup>-/-</sup> testis, we analysed testes at 18 days post partum (18dpp) during the first synchronised wave of mouse spermatogenesis. Such 18dpp wild type mouse testes contain germ cells between spermatogonia all the way through to diplotene, with 60% of cells engaged in meiotic prophase, but no post-meiotic cells (*Bellvé et al., 1977*).

Gene expression analysis of this RNAseq data (*Anders and Huber, 2010*) showed overall patterns of transcription were similar between wild type and *Rbmxl2*<sup>-/-</sup> 18dpp testes (*Figure 3A*, and

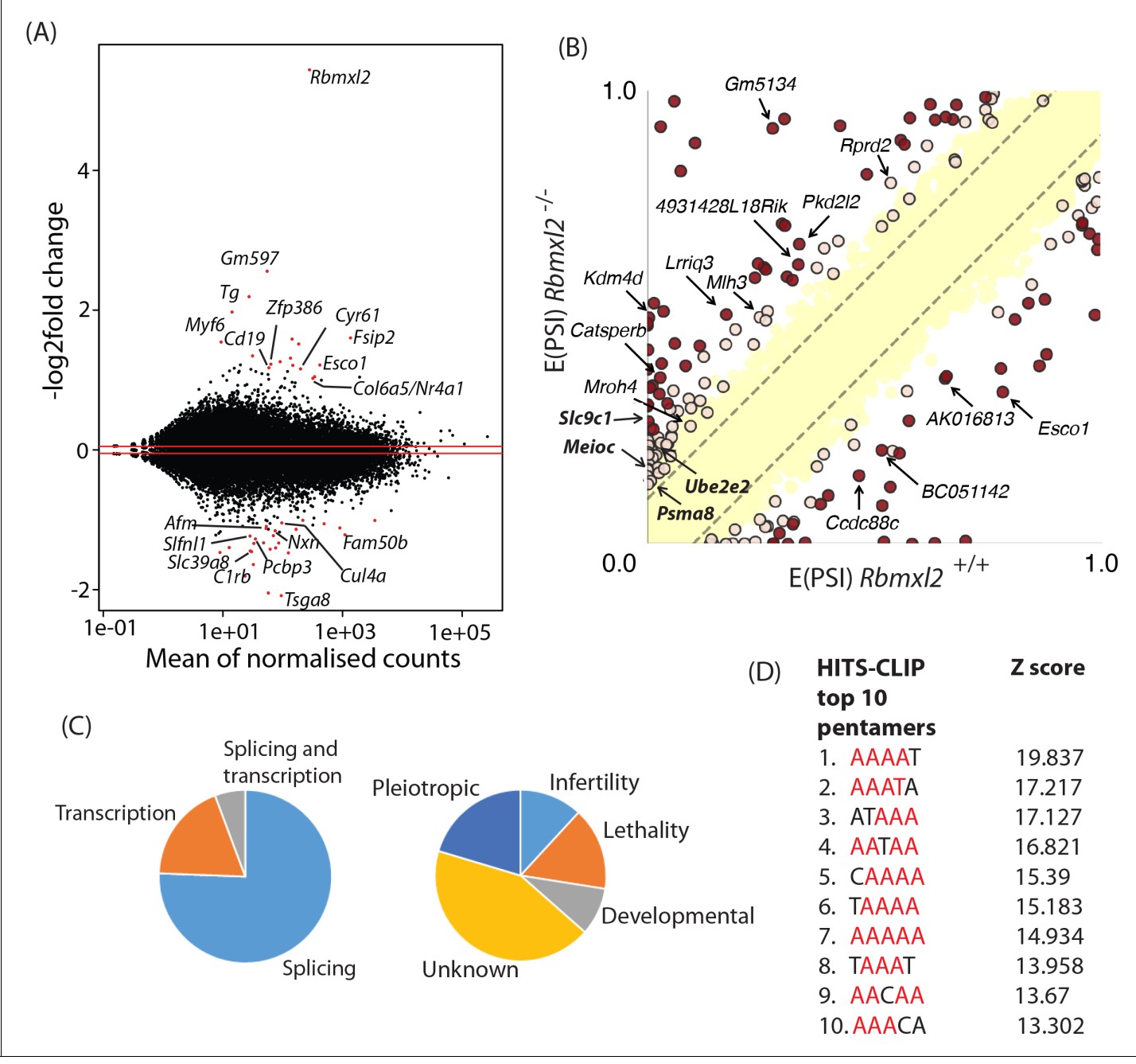

**Figure 3.** RBMXL2 protein expression controls splicing patterns of important genes during meiosis. (**A**) MAplot showing gene expression levels in 18dpp testis transcriptomes and how they change between wild type and *Rbmxl2*⁻/⁻ mice. Genes with more than a 2-fold change in gene expression, and an adjusted p value of less than 0.05 are shown as red dots. All other genes are shown as black dots. This data given in full within *Figure 3— source data 1*. (**B**) Scatterplot showing splicing changes between the wild-type and *Rbmxl2*⁻/⁻ testis detected by RNA-seq. The scatterplot shows expected percentage of exonic segment inclusion, or E(PSI) for wild-type and *Rbmxl2*⁻/⁻ testis detected by RNA-seq using the bioinformatics programme MAJIQ (*Vaquero-Garcia et al., 2016*). Splicing changes in some genes are named and arrowed. High-confidence splicing changes (P(|Δ PSI| > V)>95%) are marked in dark red for a predicted change of V = 20%, and dark yellow for a predicted change of V = 10%, and are also listed in *Figure 3—source data 2* All other quantified splice changes are in yellow (21,280 events examined). Dashed lines indicate ΔPSI of +/- 10%. (**C**) Left pie chart: Proportions of genes identified by RNAseq analysis to have defects at the splicing and transcriptional levels. Right pie chart: Proportions of phenotypes reported by whole gene knockout annotated for genes detected to have splicing defects in the absence of RBMXL2 protein (*Smith et al., 2018*) (see also *Figure 3—source data 6*). (**D**) The top 10 most frequently recovered pentamers after HITS-CLIP for RBMXL2 in the adult mouse testis (AA dinucleotides are shown in red).

DOI: https://doi.org/10.7554/eLife.39304.011

*Figure 3 continued on next page*

*Figure 3 continued*

The following source data and figure supplements are available for figure 3:

**Source data 1.** Complete list of gene expression changes between the 18dpp testes of wild type and *Rbmxl2⁻/⁻* testes detected by DESeq.
DOI: https://doi.org/10.7554/eLife.39304.015
**Source data 2.** MGI mouse phenotypes reported by the Mouse Genome Database (*Smith et al., 2018*) after whole gene knockout for RBMXL2 transcription targets.
DOI: https://doi.org/10.7554/eLife.39304.016
**Source data 3.** Complete list of 237 high-confidence, mis-regulated local splicing variations in 186 genes between wild type and *Rbmxl2⁻/⁻* testes detected by Majiq (*Vaquero-Garcia et al., 2016*).
DOI: https://doi.org/10.7554/eLife.39304.017
**Source data 4.** PCR primers used to analyse splicing events by RT-PCR that change between wild type and *Rbmxl2⁻/⁻* testes.
DOI: https://doi.org/10.7554/eLife.39304.018
**Source data 5.** Gene Ontology (GO) terms for RBMXL2 splicing targets identified.
DOI: https://doi.org/10.7554/eLife.39304.019
**Source data 6.** MGI mouse phenotypes reported after whole gene knockout for RBMXL2 splicing targets.
DOI: https://doi.org/10.7554/eLife.39304.020
**Source data 7.** List of RBMXL2 target genes that are already implicated in mouse development.
DOI: https://doi.org/10.7554/eLife.39304.021
**Figure supplement 1.** Differential expression of the *Fsip2* gene in the *Rbmxl2⁻/⁻* testis.
DOI: https://doi.org/10.7554/eLife.39304.012
**Figure supplement 2.** HITS-CLIP analysis of RBMXL2 in the adult mouse testis.
DOI: https://doi.org/10.7554/eLife.39304.013
**Figure supplement 3.** RBMXL2 CLIP and motif enrichment around regulated splicing events.
DOI: https://doi.org/10.7554/eLife.39304.014

*Figure 3—source data 1*). Only 45 genes showed a fold change greater or equal to two between the wild type and *Rbmxl2⁻/⁻* testes, with an adjusted p value of less than 0.05, and only 23 of these changes were for known protein coding genes (*Figure 3—source data 1*). The strongest difference in overall gene expression between the wild type and *Rbmxl2⁻/⁻* backgrounds was for the *Rbmxl2⁻/⁻* gene itself, as expected since the *Rbmxl2⁻/⁻* gene is deleted from the knockout mouse. We also detected strong expression changes within the *Fsip2* gene, particularly for *Fsip2* exon 16 and downstream exons that were expressed only in the wild type background (*Figure 3—figure supplement 1A and B*). Mutations in *Fsip2* correlate with defects in human sperm flagella – cellular structures which develop after meiosis (*Martinez et al., 2018*). RNAseq analysis detected more subtle expression changes in *Cul4a*, *Slc9c1* and *Tex15*, each of which are required for mouse male fertility (*Smith et al., 2018*) (*Figure 3—source data 1*) (*Kopanja et al., 2011*). Genes encoding transcription factors (*Myf6* and *Nxn*) and a signalling protein (*Cyr61*) that controls apoptosis (*Jun and Lau, 2011*) also changed expression. Mouse phenotype information at the Mouse Genome Database (MGD) (*Smith et al., 2018*) indicate that each of these latter three genes have important roles in normal development but not specifically of the testis (*Figure 3—source data 2*). Sixteen (36% of the total detected) gene expression changes in the *Rbmxl2⁻/⁻* testis were for predicted non coding RNAs of unknown function (*Figure 3—source data 1*).

## RBMXL2 protein controls splicing patterns during meiosis

We carried out further bioinformatic analysis to specifically search for mis-regulated splicing events in the *Rbmxl2⁻/⁻* testes (*Vaquero-Garcia et al., 2016*). A total of 237 high-confidence, mis-regulated local splicing variations were identified in 186 genes (*Figure 3B*, and *Figure 3—source data 3*). Using RT-PCR to distinguish splice isoforms 27 of these splicing changes were experimentally tested, validating 23/27 splice isoform switches (*Figure 3—source data 4*). Some genes had more than one splicing event controlled by RBMXL2 (e.g. the *Catsperb* gene had three events). Gene Ontology (GO) analysis showed that a number of the genes regulated at the splicing level by RBMXL2 have established roles in spermatogenesis, meiosis and germ cell development (*Figure 3—source data 5*). However, amongst the complete set of regulated genes there was no significant enrichment of particular GO terms. This is consistent with RBMXL2 regulating splicing of a functionally diverse group of genes. Analysis of knockout phenotypes provided by the MGD (*Smith et al., 2018*)

indicated that whole gene deletion of 25/186 RBMXL2-regulated genes cause male infertility. These latter target genes must thus have a key role in germ cell development (*Figure 3C*, *Figure 3— source data 6*, *Figure 3—source data 7*). Genetic deletion of some other RBMXL2 target genes cause either embryonic or neonatal lethality (33/186 genes), developmental (19/36 genes) or pleiotropic defects (43/186 genes). Cell type mis-splicing of these latter genes during meiosis could thus cause severe phenotypic effects on germ cell biology.

Fourteen (33%) of the genes originally identified to have changed overall expression levels between wild type and *Rbmxl2*[-/-] testes also changed splicing patterns (*Figure 3C*, and *Figure 3— source datas 1* and *3*). These included splice variants in *Esco1* (encoding a protein involved in chromatid cohesion), *Slc39a8* (that encodes the transporter protein responsible for cadmium toxicity in the testis) (*Dalton et al., 2005*); and the *Slc9c1* gene (annotated on the MGD as essential for male fertility [*Smith et al., 2018*]).

The RNA binding specificity of RBMXL2 protein was unknown. Thus we used high throughput sequencing cross linking immunoprecipation (HITS-CLIP) to enable us to correlate splicing changes detected within the *Rbmxl2*[-/-] 18dpp mouse testis with global RBMXL2 protein-RNA interactions (*Grellscheid et al., 2011*). Antibodies specific to mouse RBMXL2 immuno-precipitated a radiolabelled RNA protein adduct of the known size of RBMXL2 protein (50 KDa) after treatment with high concentrations of RNase (*Figure 3—figure supplement 2A*) (*Elliott et al., 2000b*). Lower concentrations of RNase were used to retrieve an average tag length of 40 nucleotides. Enriched motif analysis of the sequenced RBMXL2 CLIP tags showed that each of the top 10 5-mers contained the dinucleotide AA (*Figure 3D*). Interestingly, AA is also the dinucleotide bound by RBMX (*Moursy et al., 2014*), which is consistent with over 90% shared sequence identity within the RRMs of these proteins (*Figure 1—figure supplement 1*). These intragenic cross-linked sites mapped to the mouse genome most frequently within introns, consistent with RBMXL2 being involved in nuclear RNA-processing events (*Figure 3—figure supplement 2B*). Next we searched for CLIP tags mapping to regions near the splicing events controlled by RBMXL2, which would suggest direct regulation. We observed enrichment of RBMXL2 binding both within alternative exon sequences and in regions proximal to regulated splice sites compared to non-regulated events (*Figure 3—figure supplement 3A*). Consistent with these enriched binding occurrences, motif maps of the top pentamers identified by CLIP (*Figure 3D*) showed regions of enrichment proximal to regulated versus non-regulated splice junctions (*Figure 3—figure supplement 3B*).

## RBMXL2 protein represses splicing of exons that would compromise meiotic gene expression

Overall, the above bioinformatics analysis indicated an accumulation of defective splice isoform patterns in the *Rbmxl2*[-/-] testis. MAJIQ captures local splicing variations (LSVs) that involve both known and un-annotated (de-novo) splice sites, junctions and exons. These LSVs can correspond to both classical binary splicing events and more complex events involving three or more junctions. Since classical binary events (e.g. skipped exons) are easier to inspect and visualise with the RNAseq reads on the UCSC genome browser, we focused on the binary events for further investigation. This showed that 60% of the 87 most easily visualised classical splicing events controlled by RBMXL2 involve the altered selection of de-novo splice sites (defined as not currently annotated on the most recent build of the *Mus musculus* genome GRCm38/mm10, *Figure 3—source data 3*). These de-novo splicing variations controlled by RBMXL2 are putatively cryptic events as they insert novel internal exons that were flanked by consensus GT-AG 5′ and 3′ splice sites. Moreover, these de-novo splice sites showed low expected percent spliced in (E(PSI)) values in wild type 18 dpp testes and across a panel of twelve mouse tissues, further suggesting they are cryptic events only included in the absence of RBMXL2 and not in wild type mice (*Figure 4—figure supplement 1*). Detailed investigation indicated 84% of such cryptic exons were either not multiples of three or introduced stop codons into their mRNAs, meaning their insertion into mRNAs would disrupt protein reading frames and interfere with meiotic protein expression.

Splicing inclusion of cryptic terminal exons could also severely impact patterns of meiotic gene expression. We detected high inclusion of a cryptic terminal exon within the 5′ UTR of the *Kdm4d* gene (*Figure 4A and B*, note also increased splicing of an already annotated upstream *Kdm4d* alternative exon within the *Rbmxl2*[-/-] testis). We confirmed splicing inclusion of this *Kdm4d* cryptic exon in the *Rbmxl2*[-/-] testes using RT-PCR (primer positions are shown in *Figure 4A*). *Kdm4d* encodes a

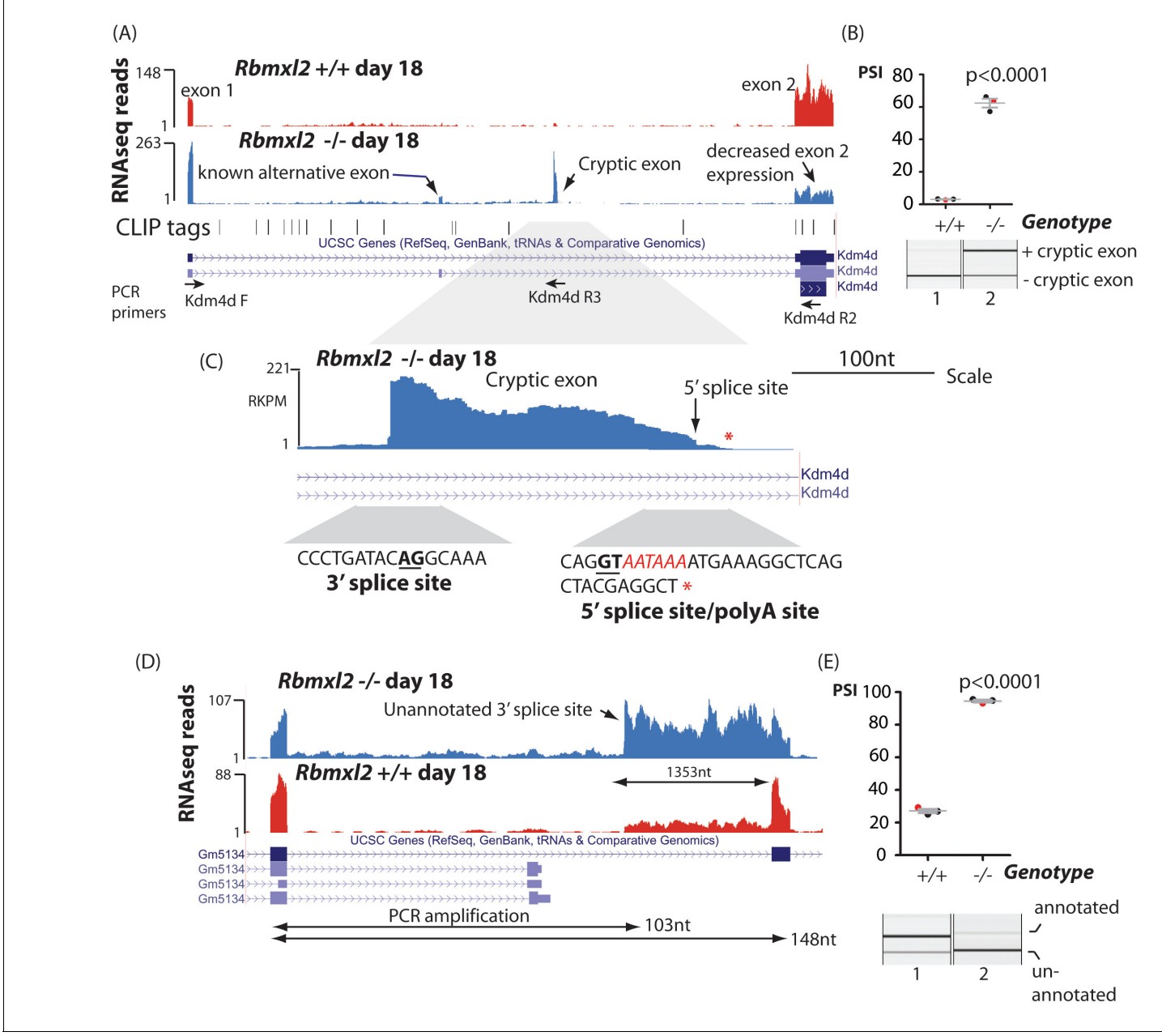

**Figure 4.** RBMXL2 protein represses splicing inclusion of exons that would compromise gene expression patterns. (**A**) Screenshot from the UCSC genome browser (***Karolchik et al., 2014***) showing high inclusion levels for a novel cryptic exon in the *Kdm4d* gene in the *Rbmxl2*-/- testis transcriptome. (**B**) Experimental confirmation of *Kdm4d* splicing changes using RT-PCR and capillary gel electrophoresis (lower panel), and data from 3 *Rbmxl2*-/- and 3 wild type 18dpp mouse testes. Individual samples shown in the capillary gel electrophoretogram are indicated as red dots. The mean PSI is shown as a grey bar with SEM as error bar. The p value was calculated using a t test. (**C**) Higher resolution screenshot (***Karolchik et al., 2014***) of the *Rbmxl2*-/- testis transcriptome showing RNAseq read density in the genomic region of the *Kdm4d* cryptic exon. The position of the cryptic 5′ splice site is indicated by a red arrow. Splice junction sequences are shown underneath with the consensus polyadenylation site AATAAA in red. Some RNAseq reads terminate downstream of the *Kdm4d* cryptic exon 5′ splice site. The final RNAseq read coverage detected downstream of this exon is shown as a red asterix. (**D**) UCSC genome browser screenshot (***Karolchik et al., 2014***) showing increased splicing selection of an cryptic 3′ splice site within the *Gm5134* mRNA in the *Rbmxl2*-/- testis transcriptome. (**E**) Chart showing experimental confirmation of the splicing switch in the *Gm5134* gene in 3 *Rbmxl2*-/- and 3 wild type 18dpp mouse testes. The individual samples shown in the capillary gel electrophoretogram are shown in the chart as red dots.

DOI: https://doi.org/10.7554/eLife.39304.022

The following figure supplements are available for figure 4:

**Figure supplement 1.** Distribution of putative cryptic splice sites across a panel of mouse tissue samples.

*Figure 4 continued on next page*

*Figure 4 continued*

DOI: https://doi.org/10.7554/eLife.39304.023

**Figure supplement 2.** (A) Sashimi plots of *Kdm4d gene in the Rbmxl2⁻/⁻ and Rbmxl2 ⁺/⁺* mice.

DOI: https://doi.org/10.7554/eLife.39304.024

**Figure supplement 3.** Examples of genes that select upstream terminal exons in the Rbmxl2 -/- testis.

DOI: https://doi.org/10.7554/eLife.39304.025

histone demethylase protein that is important for normal patterns of germ cell apoptosis in the testis (*Iwamori et al., 2011*). Exon two contains the entire CDS (Coding DNA Sequence, *Figure 4A*) of the *Kdm4d* gene. Analysis of exon junction read numbers using Sashimi plots confirmed different nuclear processing pathways are used for *Kdm4d* in wild type and knockout testes (*Figure 4—figure supplement 2A*). There were 14-fold more exon junction reads connecting *Kdm4d* exon one to the cryptic exon (140 reads), when compared to exon junction reads joining the cryptic exon to *Kdm4d* coding exon 2. This pattern is consistent with splicing inclusion of the cryptic exon being connected with use of an associated polyA site. Consistent with this, some individual RNAseq reads extended past the 5′ splice site of the *Kdm4d* cryptic exon and then terminated downstream following a consensus polyadenylation site (AATAAA) sequence (*Figure 4C*). Both bioinformatics analysis (*Figure 3—source data 1*) and qPCR (*Figure 4—figure supplement 2B*) detected reduced *Kdm4d* gene expression levels in each of 3 replicate *Rbmxl2⁻/⁻* testes compared to their wild type equivalents, but these trends were not statistically significant due to biological variability between individual mice. We also detected increased selection in the *Rbmxl2⁻/⁻* testes of another 17 splicing events that mapped to terminal exons (*Figure 3—source data 3*). These included an already annotated upstream terminal exon in the *Lrrcc1* gene (annotated on the MGD as essential for mouse fertility) (*Smith et al., 2018*); and a terminal exon in the *Slc39a8* gene (*Figure 4—figure supplement 3A and B*).

In addition to splicing of entire cryptic exons, loss of RBMXL2 protein also activated splicing selection of individual cryptic 5′ and 3′ splice sites within introns that increased the length of already annotated exons (*Figure 3—source data 3*). This kind of changed splicing pattern was observed for the *Gm5134* mRNA that encodes a solute transporter important for normal metabolism (*Figure 4D and E*). The aberrant splice isoform of the *Gm5134* transcript made in the *Rbmxl2⁻/⁻* testis uses a cryptic 3′ splice site within intron 10, increasing the size of exon 11 by 1353nt.

## RBMXL2 protein is important to prevent mis-splicing of long exons in key genes during meiosis

As well as cryptic splice sites within introns, some exon-located cryptic splice sites were activated within the *Rbmxl2⁻/⁻* testes, including within genes important for testis development (*Figure 3—source data 6*). Amongst the genes known to be important for meiosis, we observed a dip in RNA-seq density for the interior of *Meioc (meiosis specific gene with coiled coil domain)* exon five in the *Rbmxl2⁻/⁻* testis transcriptome (*Figure 5A*) (*Abby et al., 2016*; *Soh et al., 2017*). This dip in RNAseq read density was caused by high levels of cryptic splice site activation within *Meioc* exon 5. Sashimi plots showed that in the *Rbmxl2⁻/⁻* testes ∼ 28% of the splice junctions joined *Meioc* exon four to an exon-internal cryptic 3′ splice site within *Meioc* exon 5 (*Figure 5—figure supplement 1A*). A further cryptic 5′ splice site within *Meioc* exon five was also used in 22% of the *Rbmxl2⁻/⁻* testis transcripts. Use of both cryptic splice sites converted *Meioc* exon five into an exitron (an exon with internal splice sites) (*Marquez et al., 2015*; *Sibley et al., 2016*). RT-PCR analysis confirmed both these defective *Meioc* exon five splicing patterns within the *Rbmxl2⁻/⁻* testes (*Figure 5B–C*).

The cryptic splice sites within Meioc exon five are hardly ever used in the wild type 18dpp testis transcriptome. In silico analysis showed that the internal cryptic 5′ splice site within *Meioc* exon five is weak compared to 5′ splice sites for other known alternative exons (7.5th percentile by SROOGLE, compared to 57th percentile for the downstream annotated 3′ splice site (*Figure 5—figure supplement 1B*) (*Schwartz et al., 2009*). The first two nucleotide positions downstream of the *Meioc* exon 5- cryptic internal 5′ splice site are GC, so diverge from the GT consensus which is found flanking almost all eukaryotic introns. Similarly, the cryptic internal 3′ splice site within *Meioc* exon five is relatively weak (8th percentile of the average 3′ splice site strength for an alternative exon, compared to 41st percentile for the upstream annotated 3′ splice site). *Meioc* exon five is also directly bound by

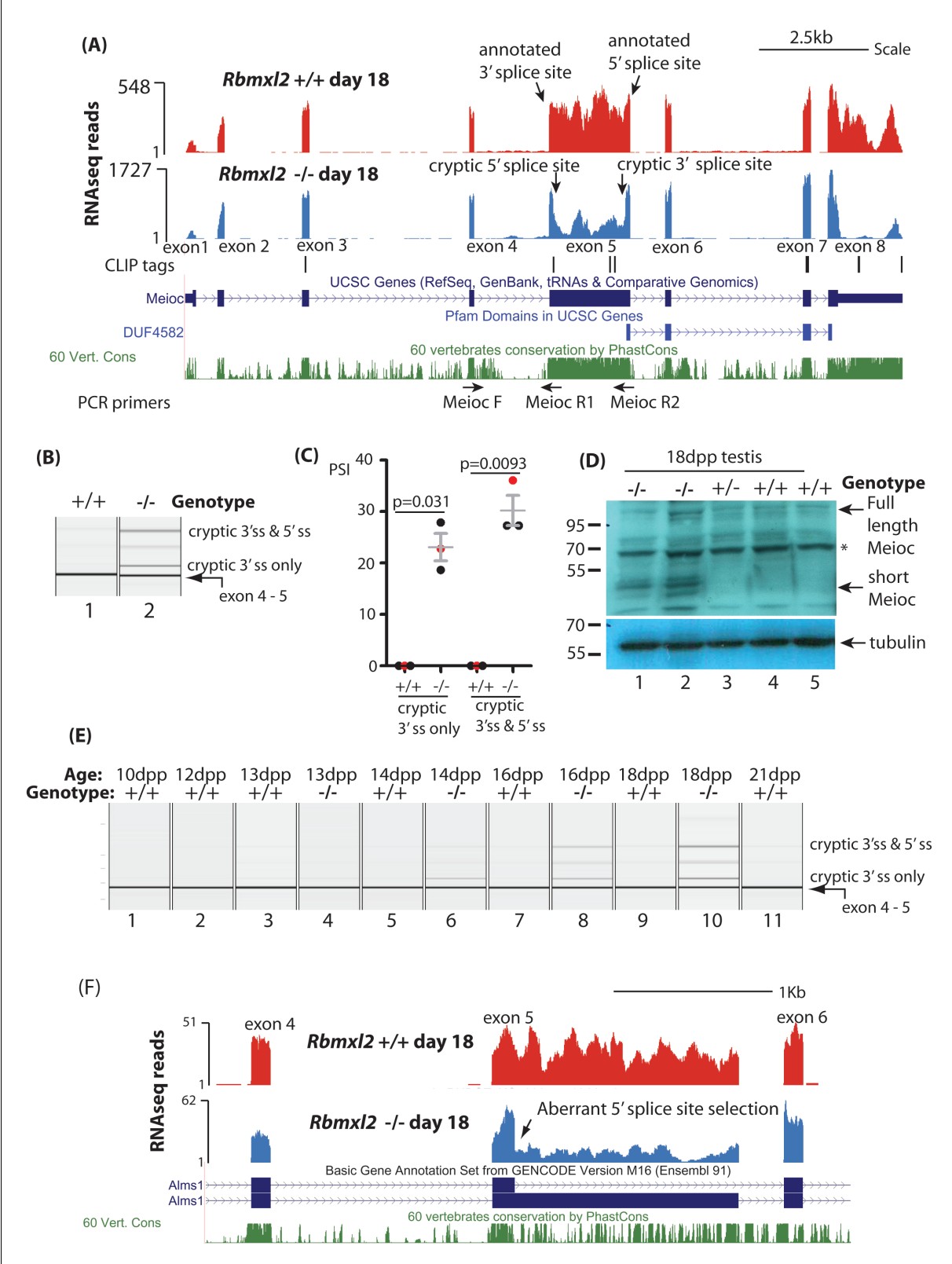

**Figure 5.** RBMXL2 protein is important to prevent splicing of long exons becoming disrupted during meiosis. (**A**) Screenshot from the UCSC genome browser (*Karolchik et al., 2014*) showing splicing pattern of *Meioc* exon five in wild type and *Rbmxl2*[-/-] testis. RNAseq read density from *Rbmxl2*[-/-] testis are shown in blue; RNAseq read density from wild type testis in red; the genomic positions of HITS-CLIP tags are mapped underneath. Splice sites for exon five are arrowed, and the binding sites for RT-PCR primers are shown. (**B**) Representative capillary gel electrophoretogram showing RT-

*Figure 5 continued on next page*

Figure 5 continued

PCR validation of the changing *Meioc* splicing pattern. This RT-PCR analysis used three primers in multiplex (MeiocF- MeiocR1 detects splicing of exon 4 to exon 5, and MeiocF-MeiocR2 detects splicing from exon four to the cryptic 3′ splice site within exon 5, or splicing of exon 4 to exon five after complete exitron removal. Detection of full length *Meioc* exon five inclusion by this latter primer combination is precluded by the large size of the amplicon. (C) Chart showing percentage splicing inclusion (PSI) of different splice variants of *Meioc* exon 5. PSI data were generated by RT-PCR from three wild type (WT) and 3 *Rbmxl2*$^{-/-}$ mice, with the individual samples shown in the capillary gel electrophoretogram in part (B) indicated as red dots. Error bars represent the SEM, and p values were calculated using t tests. (D) Western blot of 18dpp mouse testes protein probed with an antibody specific to Meioc protein (**Abby et al., 2016**). A non-specific protein that was also detected in mouse spleen as well as 18dpp testis is indicated by an asterisk. (E) Capillary gel electrophoretogram showing patterns of *Meioc* exon 5 splicing between 10dpp and 21dpp in the first wave of mouse spermatogenesis. (F) UCSC screenshot (**Karolchik et al., 2014**) showing that splicing of *Alms1* exon five is disrupted in *Rbmxl2*$^{-/-}$ testes via selection of an internal 5′ splice site.

DOI: https://doi.org/10.7554/eLife.39304.026

The following figure supplements are available for figure 5:

**Figure supplement 1.** Further splicing analysis of the Meioc gene.
DOI: https://doi.org/10.7554/eLife.39304.027

**Figure supplement 2.** Further splicing analysis of the Meioc gene.
DOI: https://doi.org/10.7554/eLife.39304.028

RBMXL2 protein, evidenced by three internal HITS-CLIP tags (**Figure 5A**). These include a CLIP tag directly overlapping the *Meioc* exon 5 internal cryptic 5′ splice site, as well as CLIP tags upstream of the internal 3′ splice site.

*Meioc* exon five is highly conserved across species suggesting it encodes a functionally important part of Meioc protein (**Figure 5A**). Exitrons frequently modify protein coding capacity of mRNAs by removing peptide coding segments from mRNAs (**Marquez et al., 2015**; **Sibley et al., 2016**). Both the cryptic splicing events in Meioc exon five remove RNA segments within the *Meioc* CDS that are multiples of 3. These events shorten the CDS just upstream of but largely not including the coding information for the MEIOC domain itself (http://pfam.xfam.org/family/PF15189). We thus predicted that a shorter Meioc protein isoform may be produced in the *Rbmxl2*$^{-/-}$ testis. Consistent with this, Western blotting with a Meioc specific antiserum (**Abby et al., 2016**) detected a 40–50 KDa Meioc protein within the *Rbmxl2*$^{-/-}$ testis, but not in wild type or heterozygote *Rbmxl2*$^{+}$ testes (**Figure 5D**). This 40–50 KDa molecular weight corresponds to an expected Meioc protein size following complete exitron removal. Although shorter Meioc protein isoforms were only detected in the *Rbmxl2*$^{-/-}$ testes, in each genotype we detected a protein doublet just above 100 KDa corresponding to the reported size of full length Meioc protein (**Abby et al., 2016**) (**Figure 5D**).

We further analysed splicing patterns during the first wave of mouse spermatogenesis to test whether appearance of the *Meioc* splicing defects corresponded with the known expression window of RBMXL2 protein in meiotic prophase (**Ehrmann et al., 2008**). Aberrant splice isoforms of *Meioc* exon five only appeared within *Rbmxl2*$^{-/-}$ testes from 14dpp, and were barely visible at any time point within similar age wild type testes. We performed similar experiments to analyse the dynamics of *Brca2* splicing in the first wave of mouse spermatogenesis. The RNAseq data predicted utilisation of an upstream cryptic 5′splice site within the 4809 nucleotide *Brca2* exon 11 (**Figure 3—source data 3**) that removes 1263 nucleotides from *Brca2* exon 11 in the *Rbmxl2*$^{-/-}$ testes (**Figure 5—figure supplement 2A**). RT-PCR analysis during the first wave of spermatogenesis showed *Brca2* exon 11 splicing was normal at 13dpp in the *Rbmxl2*$^{-/-}$ testes, with defects appearing only during and after 14dpp (**Figure 5—figure supplement 2B**). These time-course data indicate that splicing defects in both *Meioc* and *Brca2* appear and become progressively worse during meiotic prophase in the *Rbmxl2*$^{-/-}$ testes within genes that are expressed and processed normally in earlier germ cell types. An implication of this is that the aberrant splice isoforms detected in 18dpp whole testis RNA must be significantly diluted by signals from germ cells from earlier developmental stages.

Other unusually long exons in genes important for germ cell development were also mis-spliced in the *Rbmxl2*$^{-/-}$ testes including exons in the *Alms1* and *Esco1* genes (**Figure 3—source data 3**). The *Alms1* gene encodes a protein involved in microtubule organisation during cell division, and its knockout disrupts spermatogenesis resulting in a small testis phenotype (**Smith et al., 2018**). In the *Rbmxl2*$^{-/-}$ testis an upstream 5′splice site is selected within *Alms1* exon 5, which is is unusually long

at 1546nt (*Figure 5F*). This splice switch removes 1407 nucleotides from the *Alms1* mRNA, and coding information for 469 amino acids from the predicted *Alms1* reading frame.

## Ectopic expression of RBMXL2 protein represses cryptic exon splicing in vitro

The above data indicate that loss of RBMXL2 protein within spermatocytes activates utilisation of cryptic splice sites. We next tested if cryptic splice site selection would be reciprocally repressed by ectopic expression of RBMXL2 protein. The genomic region spanning the *Kdm4d* cryptic exon and

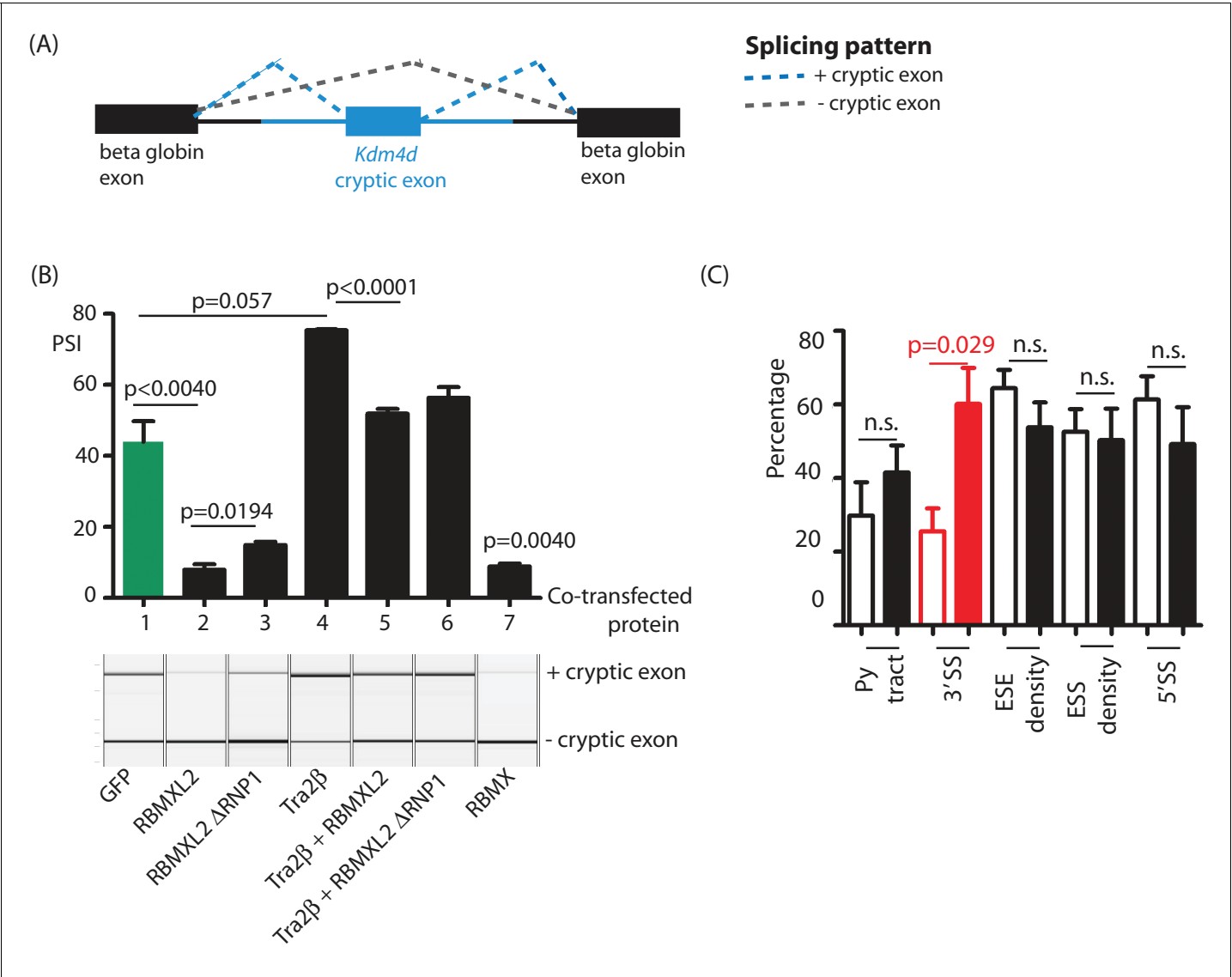

**Figure 6.** Ectopic expression of RBMXL2 protein represses cryptic exon splicing in vitro. (**A**) Schematic of the *Kdm4d* minigene, in which the *Kdm4d* cryptic exon and flanking introns (shown in blue) are inserted between two beta globin exons in the vector pXJ41 (*Bourgeois et al., 1999*). (**B**) Bar chart (upper panel) showing data from biological triplicate experiments after quantitation using capillary gel electrophoresis (n = 3 for each transfection). Capillary gel electrophoretogram (lower panel) showing splicing patterns in HEK293 cells. The minigene containing the *Kdm4d* cryptic exon and flanking intron sequences were transfected with expression plasmids encoding either GFP, RBMXL2, RBMXL2 ΔRNP1, Tra2β, RBMX or combinations of these. Splicing patterns were analysed using RT-PCR. (**C**) Bar chart showing average annotated and cryptic splice site strengths from a panel of internal cryptic exons that were activated in *Rbmxl2*[-/-] testes. Splice site strengths were quantitated as a percentile of the average value for an alternative exon using Sroogle (*Schwartz et al., 2009*). Open bars show percentile values for cryptic exons, shaded bars (black or red) show percentile values for downstream exons. Error bars represent the SEM, and p values were calculated using t tests.

DOI: https://doi.org/10.7554/eLife.39304.029

its flanking intron sequences were cloned into a minigene (*Figure 6A*). We detected ~40% splicing inclusion of the cryptic *Kdm4d* exon after co-transfection of this minigene into HEK293 cells (that do not normally express RBMXL2) with an expression vector encoding GFP (*Figure 6B* lane 1). Levels of *Kdm4d* cryptic exon splicing inclusion were repressed 6-fold upon co-transfection of a construct encoding an RBMXL2-GFP fusion protein. We also tested the activity of an RBMXL2 fusion protein without the RNP1 motif of the RRM (RBMXL2 ΔRNP1). Consistent with RNA-protein contacts not being critical for splicing repression of this cryptic exon, RBMXL2 ΔRNP1 could still repress *Kdm4d* cryptic exon splicing, although slightly less efficiently than the full length RBMXL2 fusion protein. Interestingly, splicing of the *Kdm4d* cryptic exon was also efficiently silenced by ectopic co-expression of an RBMX-GFP fusion protein (*Figure 6B* lane 7).

The above results show that RBMXL2 represses *Kdm4d* cryptic exon splicing, and is consistent with RBMXL2 providing a direct functional replacement for RBMX during meiosis. iCLIP experiments in adult mouse testis further identified the *Kdm4d* cryptic exon as directly binding the splicing activator protein Tra2β (Dalgliesh and Elliott, unpublished data). Consistent with this, Tra2β over-expression in HEK293 cells strongly increased *Kdm4d* cryptic exon splicing inclusion (*Figure 6B* lane 4). Ectopic activation by Tra2β indicated that although this *Kdm4d* exon cryptic exon is effectively ignored by the spliceosome in wild type testis, cryptic exon splicing is activated in response to increased expression of a splicing activator protein. Importantly, splicing activation by Tra2β was efficiently supressed when the minigene was co-transfected at the same time with both RBMXL2 and Tra2β (*Figure 6B*, lanes 5 and 6).

We addressed if other cryptic exons repressed by RBMXL2 might also possess the capacity to be activated by splicing regulators that bind to their associated splicing enhancer sequences. In silico analysis (*Schwartz et al., 2009*) of a panel of cryptic exons that were activated in the *Rbmxl2*[-/-] testes showed that these exons had significantly weaker 3′ splice sites, but similar Exonic Splicing Enhancer (ESE) sequence content compared to their immediately adjacent downstream exons (*Figure 6C*). These other cryptic exons that are repressed by RBMXL2 may be likewise poised for splicing during meiosis, because they bind to splicing activator proteins such as Tra2β.

## Discussion

This study finds that loss of the ancient RBMXL2 protein completely prevents sperm production in the mouse, largely because of a developmental block during meiosis. *Rbmxl2*[-/-] mouse germ cells develop as far as meiosis, thus escaping the major developmental checkpoints that operate during meiotic pachytene (*de Rooij et al., 2003*), but then undergo cell death by apoptosis after reaching diplotene. Our data further show that RBMXL2 protein is important to protect the meiotic transcriptome from a spectrum of splicing defects (*Figure 7A*). These include classic manifestations of cryptic splicing that would be strongly deleterious to gene expression (*Marquez et al., 2015*; *Sibley et al., 2016*), such as insertion of novel cryptic exons and premature polyadenylation sites that would disrupt protein coding sequences, and the selection of cryptic splice sites that would shorten or extend exon lengths. Previous work reporting poor conservation of splicing patterns between species suggested that alternative splicing might not control fundamental aspects of germ cell biology (*Kan et al., 2005*). In contrast, the work presented here suggest a generic requirement for cryptic splice site suppression during meiosis that could explain the conservation of an *Rbmxl2* gene in all placental mammals, even if individual cryptic splice sites are not conserved between species' genomes (*Elliott et al., 2000b*).

Cells in meiotic prophase might be particularly susceptible to cryptic splice site poisoning because of their high levels of transcription, altered splicing regulator expression and relaxed chromatin environments (*de la Grange et al., 2010*; *Licatalosi, 2016*; *Soumillon et al., 2013*; *Pan et al., 2008*; *Wang et al., 2008*; *Clark et al., 2007*; *Grosso et al., 2008*; *Yeo et al., 2004*). In silico data suggest that the cryptic exons repressed by RBMXL2 would be poised for splicing inclusion in conditions of high concentrations of splicing activators because of their strong ESE contents. We suggest a model where RBMXL2 protein may buffer the activity of splicing activator proteins to prevent such cryptic splice site selection from happening in meiosis (*Figure 7B*). This model is also consistent with previously reported data showing that RBMXL2, RBMX and RBMY proteins physically interact with and antagonise the splicing activity of Tra2β and SR proteins (*Liu et al., 2009*; *Nasim et al., 2003*; *Venables et al., 2000*; *Elliott et al., 2000a*) (*Figure 7B*). Data presented in this study also show that

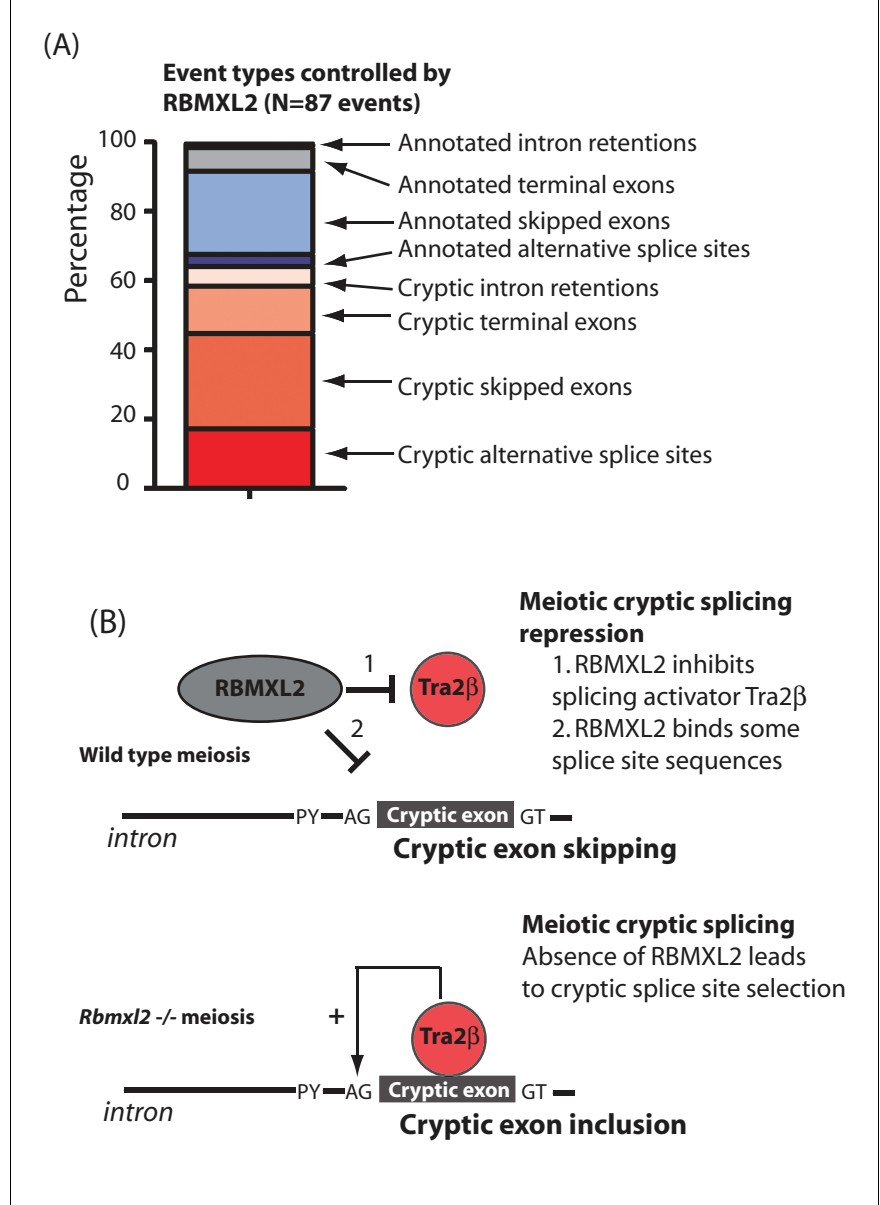

**Figure 7.** RBMXL2 protein is needed to enable production of proper splice isoform patterns during meiosis. (**A**) Stacked bar chart summarising percentage types of splicing event mis-regulated in the *Rbmxl2⁻ᐟ⁻* testes. Details of the individual genes contributing to this list are given in *Figure 3—source data 3*. (**B**) Models for RBMXL2 function. In wild type mice, RBMXL2 protein may block cryptic exon splicing through: 1. Inhibition of splicing regulator proteins including Tra2β protein via protein-protein interactions (*Venables et al., 2000*); or 2. Direct RNA protein interactions of RBMXL2 with important splicing regulatory sequences in the pre-mRNA. In the *Rbmxl2⁻ᐟ⁻* background the absence of this buffering would result in an increase in splicing inclusion of some cryptic exons which contain Tra2β binding sites.
DOI: https://doi.org/10.7554/eLife.39304.030

*Kdm4d* cryptic exon splicing activation by Tra2β is blocked in vitro by co-expression of RBMXL2. As an alternative and not mutually exclusive model, RBMXL2 protein-RNA binding may sterically block recognition of some cryptic splice sites during meiosis, including the cryptic 5′ splice site within *Meioc* exon five which was utilised in the *Rbmxl2⁻ᐟ⁻* testis (*Figures 5A* and *7B*). However, no HITS-CLIP tags for RBMXL2 were detected near the *Kdm4d* cryptic exon. Furthermore, a version of RBMXL2 deleted for the key RNP1 motif involved in RNA-protein interactions could still repress

*Kdm4d* cryptic exon splicing (albeit slightly less efficiently than the wild type RBMXL2 protein). Thus buffering protein interactions between RBMXL2 and proteins like Tra2β may be potentially more important in repressing cryptic splicing in meiosis than direct RBMXL2 RNA-protein interactions.

Loss of RBMXL2 disrupts expression of many downstream target genes in parallel, both at splicing and transcriptional levels. This makes it difficult to resolve whether the phenotype detected in the *Rbmxl2*$^{-/-}$ mouse testes is caused by aberrant expression of a single target RNA, or is a compound phenotype involving multiple genes. Some genes affected by RBMXL2 protein deletion are annotated as important for spermatogenesis on the MGD (*Smith et al., 2018*). These include *Alms1* (*Arsov et al., 2006*; *Collin et al., 2005*); *Brca2* which is required for meiotic prophase (*Sharan et al., 2004*); *Kdm4d*, which helps control apoptosis during male germ cell development (*Iwamori et al., 2011*); and *Cul4a* that encodes an important ubiquitin ligase needed for DNA damage response. Two factors complicate direct comparison of MGD phenotypes with those after RBMXL2 knockout. Firstly, conventional knockouts will report a phenotype the first time a gene is needed in a developmental pathway. In contrast RBMXL2 is only expressed from the onset of meiosis, so any defects in target gene expression that occur without RBMXL2 could result in a later phenotype. For example, *Meioc* knockout phenotype analysis has concentrated on entry into meiosis (*Abby et al., 2016*; *Soh et al., 2017*). Our data show that *Meioc* exon five is still spliced normally on meiotic entry in the *Rbmxl2*$^{-/-}$ mouse (at 12dpp), but becomes compromised later on during meiotic prophase (when RBMXL2 is expressed). It is possible that *Rbmx* and *Rbmy* will provide an equivalent function to *Rbmxl2* in spermatogonia and early spermatocytes, and that it is only when meiotic sex chromosome inactivation initiates during pachytene that germ cells will depend on *Rbmxl2*. Consistent with this, in transfected cells RBMX was able to suppress a cryptic splice exon in the *Kdm4d* with similar efficiency to the RBMXL2 protein. Secondly, the MGD phenotype data (*Smith et al., 2018*) is after gene knockout, whereas Rbmxl2 deletion changes patterns of splice isoforms.

High levels of gene expression and alternative splicing have also been observed in the nervous system, and this is another anatomic site where pathological cryptic exon inclusion has been reported. Neuronal cryptic exon inclusion occurs following depletion of the splicing repressor protein TDP43, and might contribute to neuron death in neurological diseases like ALS and Alzheimer's disease (*Sun et al., 2017*). Cryptic splicing has also been observed in cultured human cells after depletion of the splicing repressor proteins hnRNP C, hnRNPL and PTB (*Ling et al., 2016*; *McClory et al., 2018*; *Zarnack et al., 2013*), and in mouse oocytes depleted for the splicing factor SRSF3 (*Do et al., 2018*). Taking all these results into consideration we suggest that RBMXL2 has a key role controlling the meiotic transcriptome, and suggest that human male infertility caused by loss of *RBMXL2* or its paralog RBMY may be associated with germ cell type-specific cryptic splicing.

# Materials and methods

## Key resources table

| Reagent type (species) or resource | Designation | Source or reference | Identifiers | Additional information |
|---|---|---|---|---|
| Gene (M. musculus) | *Rbmxl2* | | | |
| Strain, strain background (M. musculus) | C57Bl/6 | | | |
| Genetic reagent (M. musculus) | *Rbmxl2* knockout | this publication | | |
| Genetic reagent (M. musculus) | FlpE | *Rodríguez et al., 2000* | | |
| Genetic reagent (M. musculus) | PGKcre | *Lallemand et al., 1998* | | |
| Cell line (H. sapiens) | HEK293 | | | |
| Antibody | Sheep polyclonal anti-Rbmxl2 | *Ehrmann et al. (2008)* | | (1:700) |

*Continued on next page*

*Continued*

| Reagent type (species) or resource | Designation | Source or reference | Identifiers | Additional information |
|---|---|---|---|---|
| Antibody | Guinea pig polyclonal anti-Meioc | *Abby et al. (2016)* | | (1:1000) |
| Antibody | Mouse monoclonal anti-tubulin | Sigma | catalog number T6793 | (1:1000) |
| Antibody | Rabbit polyclonal anti-GAPDH | Abgent, San Diego CA | catalog number AP7873b | (1:1000) |
| Antibody | Rabbit polyclonal anti-caspase 3 | Proteintech | catalog number 25546–1-AP | (1:100) |
| Antibody | Rabbit polyclonal anti-Sycp3 | Life span | Cat# LS-B175, RRID:AB_2197351; lot#73619 | (1:500) |
| Antibody | Mouse monoclonal anti-Sycp3 [Cor 10G11/7] | Abcam | Cat#ab97672 RRID:AB_10678841; lot#GR282926-1 | (1:500) |
| Antibody | Guinea Pig polyclonal anti-Sycp1 | Laboratory of Howard Cooke | | (1:200) |
| Antibody | Mouse monoclonal anti-phospho H2AX [JBW301] | Millipore | Cat# 05–636, RRID:AB_309864; lot#1997719 | (1:3000) |
| Antibody | Rabbit polyclonal anti-H3 trimethyl K9 | Abcam | Cat# ab8898, RRID:AB_306848; lot# GR250850-2 | (1:500) |
| Antibody | Mouse monoclonal anti-H3 phospho (Ser10) | Abcam | Cat# ab14955, RRID:AB_443110; lot# GR192274-8 | (1:1000) |
| Antibody | Human polyclonal anti-Centromere Protein | Antibodies Incorporated | Cat#15-235-0001 | (1:50) |
| Antibody | Mouse monoclonal anti-AcSmc3 | From Katsuhiko Shirahige, University of Tokyo. *Nishiyama et al. (2010)* | | (1:1000) |
| Antibody | Goat polyclonal anti-Mouse Alexa Fluor 488 | Invitrogen | Cat# A11001 | (1:500) |
| Antibody | Goat polyclonal anti-Rabbit Alexa Fluor 488 | Invitrogen | Cat#A11008 | (1:500) |
| Antibody | Goat polyclonal anti-Human Alexa Fluor 488 | Invitrogen | Cat# A11013 | (1:500) |
| Antibody | Goat polyclonal anti-Guinea Pig Alexa Fluor 488 | Invitrogen | Cat# A11073 | (1:500) |
| Antibody | Goat polyclonal anti-Mouse Alexa Fluor 594 | Invitrogen | Cat# A11005 | (1:500) |
| Antibody | Goat polyclonal anti-Rabbit Alexa Fluor 594 | Invitrogen | Cat# A11012 | (1:500) |
| Antibody | Goat polyclonal anti-Rabbit Alexa Fluor 647 | Invitrogen | Cat# A32733 | (1:500) |
| Antibody | Goat polyclonal anti-Guinea Pig Alexa Fluor 647 | Invitrogen | Cat# A21450 | (1:500) |
| Dye | DAPI (4',6-diamidino-2-phenylindole) | Biotium | Cat# 40043; lot# 15D1117 | |

*Continued on next page*

*Continued*

| Reagent type (species) or resource | Designation | Source or reference | Identifiers | Additional information |
|---|---|---|---|---|
| Recombinant DNA reagent | PXJ41 Kdm4d | this paper | | |
| Sequence-based reagent | primers | this paper | | designed using Primer3 http://primer3.ut.ee/ |
| Sequence-based reagent | qPCR primers | this paper | | designed using Primer3 http://primer3.ut.ee/ |
| Commercial assay or kit | Rnaeasy plus kit | Qiagen | catalog number 74134 | |
| Commercial assay or kit | DNA free | Ambion | catalog number AM1906 | |
| Commercial assay or kit | Superscript Vilo | Invitrogen | catalog number 11754 | |
| Chemical compound, drug | Bouins | Sigma | catalog number HT10132 | |
| Software, algorithm | Graphpad prism | https://graphpad.com | | |
| Software, algorithm | Sroogle | http://sroogle.tau.ac.il/ | | |

## Generation of KO mice

A targeting construct in which the *Rbmxl2* open reading frame was flanked by *LoxP* sites was made using standard molecular biology techniques, and electroporated into ES129 cells. Positive clones were injected into blastocysts to create chimaeras, and bred to yield agouti pups heterozygous for the targeted locus (Ozgene, Perth, Australia). The original mice containing the *Neomycin* gene (*Figure 1—figure supplement 1A*) were crossed to *FlpE* mice to remove the *Neo* gene and to generate the *Rbmxl2* $^{LoxP}$ conditional allele (*Figure 1—figure supplement 1B*). Mice containing the *Rbmxl2* $^{LoxP}$ conditional allele were crossed with mice expressing *PGKCre*, resulting in deletion of the *Rbmxl2* reading frame (*Figure 1—figure supplement 1C*). Genetic structures of the wild type and targeted alleles were confirmed by Southern blotting. Genomic DNA was prepared from mouse tails, cut with *BamHI*, run on an agarose gel and blotted onto a Hybond N nylon membrane (GE Healthcare, Little Chalfont, UK). Blots were probed using an internal probe Enp (generated using primers enpF 5'-ACTGTTGATTCCCCTTCCAAC-3' and enp R 5'- ACTCCTGCCTGTGATTGGTC-3', and α- $^{32}$ P labelled by random priming). The correct insertion of the targeting vector in the genome was confirmed by cutting genomic DNA with *PshA1* and *EcoRV* and hybridizing with 5' (generated from genomic DNA using 5'-agcattcagcaaaggctcac-3' and 5'- ttaaaactgagggagactgc-3') and 3' (generated from genomic DNA using 5'-actgcatagttgtagccatc-3' and 5'- tgcattctctttaggctcatttc-3') probes respectively.

## Animal work

Animal research was carried with the approval of the Newcastle University animal research ethics committee and the UK Government Home Office (Home Office project Licence Number PIL 60/4455).

## Analysis of mouse germ cell development in vivo

Testis/body weight ratios, and sperm counts were measured on a C57Bl6 background. In order to determine sperm counts, the cauda epididymis was dissected in PBS and the sperm were counted in a haemocytometer.

## Preparation of RNA for RNA-Seq

Mice were back crossed onto the C57Bl6 background for eight generations, and we used back crossed male mice for subsequent analysis. Testes were homogenized in RLTplus buffer from Qiagen before purification with the Rneasy plus kit (Qiagen, 74134). The RNA was then re-purified with the kit before Dnase digestion (Ambion).

## RNA-Seq analysis

Paired-end sequencing was done for six samples in total (three biological replicates of 18dpp wild-type and Rbmxl2⁻/⁻ testis) using 75 bp reads. Libraries were prepared using TruSeq Stranded mRNA Library Prep Kit (Illumina) and sequenced; 75 bp single reads were sequenced on a NextSeq 500 (Illumina, using the mid output v2 150 cycles kit). The base quality of the raw sequencing reads were checked using FastQC. Trimmomatic (v0.32) was used to remove adapters and to trim the first twelve bases and the last base at position 76 (*Bolger et al., 2014*). Reads were aligned to the UCSC D3c. 2011 (GRCm38/mm10) assembly of the mouse genome using STAR (v2.5.2b) (*Dobin et al., 2013*). Alternative splicing events were assessed using MAJIQ and VOILA software packages (*Vaquero-Garcia et al., 2016*). Briefly, uniquely mapped, junction-spanning reads were used by MAJIQ to construct splice graphs for transcripts from a custom Ensembl transcriptome annotation and to quantify PSI (within conditions) and ΔPSI (between conditions) for all local splicing variations (LSVs). The captured LSVs include classical alternative splicing events (e.g. cassette exons, alternative 5' splice sites, etc.) as well as more complex variations (*Vaquero-Garcia et al., 2016*). LSVs with an expected change of greater than 10% were then visualized using VOILA to produce splice graphs, violin plots representing PSI and ΔPSI quantifications, and interactive HTML outputs for changes between wild type and Rbmxl2⁻/⁻.

## In silico analysis of mouse cryptic exons

Splicing patterns of identified target genes were analysed using the UCSC and IGV mouse genome browsers (*Karolchik et al., 2014*; *Thorvaldsdóttir et al., 2013*). A panel of 10 cryptic exons from the *Catsperb* (two exons), *Ccdc60*, *Kdm4d*, *Lrrcc63*, *4933409G03Rik*, *Tcte2*, *Ube2e2*, *Fam178b* genes) and their immediately downstream exons were monitored for strength of splicing sequences using Sroogle (*Schwartz et al., 2009*). A panel of 19 exons were analysed for length and translation termination codons content (*1700074P13Rik*, *4833439L19Rik*, *ccdc60*, *Dnah12*, *Hmga2*, *Map2k5*, *Rbks*, *Tcte2*, *Ube2e2*, *4933409G03Rik*, *4930500J02Rik*, *4932414N04Rik*, *Fam178b*, *Ccdc144b*, *Ston1*, *Usp54*, *Lca5l*, and *Catsperb* and *Lca5l*).

## Detection of splice isoforms in mouse testis RNA

The levels of different mRNA splice forms were detected in DNAse treated total RNA isolated from 18dpp mouse testes. cDNA was synthesised using superscript Vilo (Invitrogen) and PCR amplified, followed by either capillary gel electrophoresis (Qiaxcel) or agarose gel electrophoresis (*Grellscheid et al., 2011*). Gene-specific primers specific for different splice isoforms are given in *Figure 3—source data 4*.

## Hits-clip

HITS-CLIP was performed as previously described (*Licatalosi et al., 2008*) using an antibody specific to RBMXL2 (*Ehrmann et al., 2008*). In short, for the CLIP analysis a single mouse testis was sheared in PBS and UV crosslinked. After lysis, the whole lysate was treated with DNase and RNase, followed by radiolabelling and linker ligation. After immunoprecipitation with purified antisera specific to RBMXL2 (*Ehrmann et al., 2008*), RNA bound RBMXL2 was separated on SDS-PAGE. A thin band at the size of 65 kDa (RBMXL2 migrates at around 50 kDa and MW of 50 nt RNA is about 15 kDa) was cut out and subject to protein digestion. RNA was recovered and subject to Illumina sequencing, and the reads were aligned to the mouse genome. HITS-CLIP tags mapped to 22,438 protein coding genes. HITS-CLIP data were further analysed to extract most frequently occurring pentamer nucleotide sequences within genes (ncRNA, ORF, 3'UTR, 5'UTR and intron). A window between +50 and −50 of the cross link site was analysed (to avoid crosslinking bias), and after correction for average pentamer occurrence in control sequences (as described in *Wang et al., 2010*). Each pentamer was only counted once in the analysed window.

## *Kdm4d* minigene analysis

The *Kdm4d* cryptic exon and flanking intron sequences were PCR-amplified from mouse genomic DNA using the cloning primers Kdm4dF (5'-AAAAAAAAGAATTCCCACACAGCAAAACCCTCTC-3') and Kdm4dR (5'-AAAAAAAAGAATTCGCCACCTTTTGCTATTCCTTT-3'). The PCR products were digested with *EcoR1* restriction enzyme and cloned into the pXJ41 vector using the *Mfe1* site

midway through the 757 nucleotide β-globin intron (*Bourgeois et al., 1999*). Analysis of splicing patterns transcribed from minigenes was carried out in HEK293 cells as previously described (*Grellscheid et al., 2011*; *Venables et al., 2005*) using primers within the β-globin exons of pXJ41, pXJ41F 5′-GCTCCGGATCGATCCTGAGAACT-3′) and pXJ41R (5′-GCTGCAATAAACAAGTTCTGCT-3′).

### Detection of proteins in mouse testis

For protein detection by immunohistochemistry, testes were fixed in Bouins, embedded in paraffin wax, sectioned and attached to glass slides. Sections were prepared and immunohistochemistry carried out as previously described (*Grellscheid et al., 2011*). Primary antibodies were specific to caspase 3, p17-specific (Proteintech Europe Ltd). Protein detection by Western was as previously described (*Grellscheid et al., 2011*) using primary antibodies specific to RBMXL2 (*Ehrmann et al., 2008*), Meioc (*Abby et al., 2016*), and control antibodies specific to tubulin (Sigma T6793) and GAPDH (Abgent,San Diego CA, AP7873b).

### Spermatocyte spreading

The tunica was removed from testes, and seminiferous tubules macerated with razor blades in a small volume of DMEM media. DMEM was added to 4 ml, and debris allowed to settle for 10 min. Cells were then pelleted from the suspension at 233 g for 5 min. The pellet was resuspended in 1–2 ml DMEM per animal. 5 drops of 4.5% sucrose were added to the centre of each clean glass slide using a Pasteur pipette. A single drop of the cell suspension was added to each slide from a height of 15–20 cm, followed by one drop of 0.05% Triton-X-100. The slides were incubated in a humid chamber for 10 min, then eight drops of fixative (2% PFA, 0.02% SDS, pH 8) were added to each slide, and incubated in humid chamber for 20 min. Slides were dipped in water to wash and allowed to air dry before storing at −80°C (*Crichton et al., 2017*).

### Antibody staining and image capture

Slides were blocked for one hour in 0.15% BSA, 0.1% Tween-20, 5% goat serum PBS, then incubated with primary antibodies for three hours at room temperature or 4°C overnight. Primary antibodies were used at the following concentrations, diluted in block solution: rabbit anti-Sycp3 (1:500, Lifespan Cat# LS-B175 RRID:AB_2197351), mouse anti-Sycp3 (1:500, Abcam Cat# ab97672 RRID:AB_10678841), guinea pig anti-Sycp1 (1:200, from Howard Cooke), mouse anti-γH2AX (1:3000; Millipore Cat# 05–636, RRID:AB_309864), rabbit anti-H3K9me3 (1:500; Abcam Cat# ab8898, RRID:AB_306848), mouse anti-phospho H3 (1:1000; Abcam Cat# ab14955, RRID:AB_443110), human anti-centromere (1:50; Antibodies Incorporated, Cat#15-235-0001), mouse anti-AcSmc3 (1:1000, a gift from Katsuhiko Shirage [*Nishiyama et al., 2010*]). Slides were washed three times for five minutes in PBS before incubating with 1:500 secondary antibodies and ng/ul DAPI (4′,6-diamidino-2-phenylindole) for one hour at room temperature in darkness. Slides washed again three times and mounted in 90% glycerol, 10% PBS, 0.1% p-phenylenediamine.

Images were captured with Micromanager imaging software using an Axioplan II fluorescence microscope (Carl Zeiss) equipped with motorised colour filters. Centromere and H3K9me3 staining was imaged by capturing z-stacks using a piezoelectrically-driven objective mount (Physik Instrumente) controlled with Volocity software (PerkinElmer). These images were deconvolved using Volocity, and a 2D image generated in Fiji.

Nuclei were staged by immunostaining for the axial/lateral element marker SYCP3. Pachytene nuclei were identified in this analysis by complete co-localisation of SYCP3 and either the transverse filament marker SYCP1 on all nineteen autosomes, or by the presence of nineteen bold SYCP3 axes in addition to two paired or unpaired sex chromosome axes of unequal length. Asynapsed pachytene nuclei were judged to be nuclei containing at least one fully synapsed autosome, and at least one autosome unsynapsed along >50% of its length (*Crichton et al., 2017*).

### Quantification and statistical analysis

Quantification and statistical analyses were done using Graphpad (http://www.graphpad.com/).

## Spermatocyte staining analysis

All scoring of immunostained chromosome spreads was performed blind on images after randomisation of filenames by computer script. Three $Rbmxl2^{-/-}$ and three wild type control animals were analysed for each experiment. Meiotic substaging analysis involved 179 nuclei from control animals, and 138 from $Rbmxl2^{-/-}$ mice. Diplotene phospho-H3 analysis involved 128 control and 157 $Rbmxl2^{-/-}$ nuclei, and the proportion of nuclei with bold centromeric signal was analysed using a Fisher's test. Pachytene asynapsis was measured in 108 control and 92 $Rbmxl2^{-/-}$ nuclei, and the data analysed using a Fisher's test. Centromere positioning and H3K9me3 staining was assessed across 73 control and 65 $Rbmxl2^{-/-}$ pachytene nuclei, and 66 diplotene control and 66 $Rbmxl2^{-/-}$ diplotene nuclei. These data were analysed using Wilcoxon Rank Sum test.

## Acknowledgements

This work was funded by the BBSRC (grant BB/P006612/1, awarded to DJE) and the Wellcome Trust (grant 089225/Z/09/Z, awarded to DJE). JC and IRA were funded by the Medical Research Council (http://www.mrc.ac.uk/) through an intramural programme grant award to IRA (MC_PC_U127580973). Work by MRG and YB was supported by R01 AG AG046544 to YB. We thank Laura Bellutti and Gabriel Livera for kindly supplying the Meioc antiserum. We are very grateful to Donal O'Carroll (Edinburgh University), Nick Europe Finner, Julian Venables and Louise Reynard for comments on the manuscript.

## Additional information

### Funding

| Funder | Grant reference number | Author |
|---|---|---|
| Biotechnology and Biological Sciences Research Council | BB/P006612/1 | Ingrid Ehrmann<br>Yilei Liu<br>David J Elliott |
| Medical Research Council | MC_PC_U127580973 | James H Crichton<br>Ian R Adams |
| National Institutes of Health | R01 AG AG046544 | Matthew R Gazzara<br>Yoseph Barash |
| Wellcome Trust | 089225/Z/09/Z | David J Elliott |

The funders had no role in study design, data collection and interpretation, or the decision to submit the work for publication.

### Author contributions

Ingrid Ehrmann, Conceptualisation, Funding acquisition, Investigation, Writing-review and editing; James H Crichton, Ian R Adams, Yoseph Barash, Funding acquisition, Supervision, Investigation, Writing-review and editing; Matthew R Gazzara, Yilei Liu, Sushma Nagaraja Grellscheid, Dirk de Rooij, Investigation, Writing-review and editing; Katherine James, Resources, Software, Investigation, Writing-review and editing; Tomaž Curk, Data curation; Jannetta S Steyn, Simon Cockell, Visualization, Writing-review and editing; David J Elliott, Conceptualisation, Funding acquisition, Supervision, Investigation, Writing-original draft

### Author ORCIDs

Tomaž Curk http://orcid.org/0000-0003-4888-7256
Dirk de Rooij https://orcid.org/0000-0003-3932-4419
Jannetta S Steyn https://orcid.org/0000-0002-0231-9897
Ian R Adams https://orcid.org/0000-0001-8838-1271
Yoseph Barash https://orcid.org/0000-0003-3005-5048
David J Elliott http://orcid.org/0000-0002-6930-0699

## Ethics

Animal experimentation: Animal research was carried with the approval of the Newcastle University animal welfare ethical review body and the UK Government Home Office (Home Office project Licence Numbers PIL 60/4455 and PE51EB2EB).

## Decision letter and Author response

Decision letter https://doi.org/10.7554/eLife.39304.035
Author response https://doi.org/10.7554/eLife.39304.036

## Additional files

### Supplementary files

• Transparent reporting form
DOI: https://doi.org/10.7554/eLife.39304.031

### Data availability

RNAseq data are available via the Gene Expression Omnibus (GEO) using GEO accession number GSE101511

The following dataset was generated:

| Author(s) | Year | Dataset title | Dataset URL | Database and Identifier |
|---|---|---|---|---|
| Ehrmann I, Crichton J, Gazzara MR, de Rooij DG, Steyn J | 2018 | HnRNPGT knockout disrupts pre-mRNA splicing and arrests sperm development prior to meiotic metaphase | https://www.ncbi.nlm.nih.gov/geo/query/acc.cgi?acc=GSE101511 | NCBI Gene Expression Omnibus, GSE101511 |

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
