## [Decision Letter]

[Editors’ note: the authors were asked to provide a plan for revisions before the editors issued a final decision. What follows is the editors’ letter requesting such plan.]

Thank you for sending your article entitled "An ancient germ cell-specific RNA binding protein protects the germline from cryptic splice site poisoning" for peer review at *eLife*. Your article is being evaluated by a Reviewing Editor, and three reviewers and the evaluation is being overseen by a Reviewing Editor and Marianne Bronner as the Senior Editor.

Given the list of essential revisions, including new experiments, the editors and reviewers invite you to respond within the next two weeks with an action plan and timetable for the completion of the additional work. We plan to share your responses with the reviewers and then issue a binding recommendation.

In this manuscript, Ehrmann and colleagues characterize the germ cell-specific RNA binding protein hnRNP GT. The authors generated a conditional knockout mouse targeting hnRNP GT, and demonstrate significant defects in testis histology and spermatogenesis in mutant animals. Specifically, the authors demonstrate that defects in knockout animals likely stem from problems during a specific stage of meiosis. Given the observed phenotypes, transcriptome-wide analysis is performed looking both at steady-state mRNA levels as well as splicing patterns. The authors find both transcript-level and splicing-level differences between wild type and mutants, but their data suggest more prevalent defects in the latter layer of gene regulation. In particular, in addition to aberrant regulation of annotated alternative exons, the authors identify a more significant proportion of 'cryptic'/minor splice sites being utilized in mutant animals, and the use of these splice sites are predicted to have deleterious effects on encoded protein products. Additional experiments include HITS-CLIP assays that determine the sequence binding preference of hnRNP GT (motifs with AA dinucleotides) and information about direct targets of this RNA binding protein. Finally, the authors show in cell lines that over-expression of hnRNP GT has the ability to suppress usage of a cryptic splice site, even in the context of co-expression of the activating SR protein Tra2β.

Overall, this is a carefully conducted investigation that should be of considerable interest to the readership of *eLife* after suitable revision. It is requested that authors provide a proposed plan for revisions that can be performed within approximately two months, based on the comments below. In particular, it was felt that the paper would be significantly strengthened by the inclusion of additional data supporting functional roles for the isoforms highlighted by the authors, and in this regard their plan should pay special attention to point 3 below. It was also felt that the section of the paper relating to functional consequences of isoform switches could be better focused and the authors should make an effort to revise this section accordingly.

Essential revisions:

1) Subsection “hnRNP GT protein controls splicing patterns during meiosis”. The manuscript states that 23 of 27 tested splicing events were validated, citing Source Data 3. But this file has no data – only has a list of primers used for RT-PCR. It would be appropriate to include the results of the RT-PCR – as PSI (mean {plus minus} sd) for WT and KO conditions, and to define how an event was defined as validated. Did validation only require a statistically significant change in the same direction as for RNA-Seq? Ideally show the RT-PCR values alongside the MAJIQ values for PSI.

The RT-PCR examples shown in Figure 4B, Figure 5B and Figure 5—figure supplement 2A,B, look convincing and do not need quantitation to be convincing. Nevertheless, it would be good to show the PSI numbers to allow direct comparison with the MAJIQ output.

2) Many genes are identified with differences in overall transcript abundance and/or changes in alternatively spliced isoforms. The authors largely ignore the 45 genes identified with differences in RNA level between WT and KO testis. The authors should comment on these genes and provide information on what the encoded proteins do, and how changes in their levels (caused by hnRNP GT KO) might contribute to the meiotic defects observed.

3) The analysis of alternative splicing changes is not explored in sufficient depth to provide insight into why KO germ cells arrest in meiosis. The authors focus on select examples of genes previously shown to be important for spermatogenesis and that also express alternatively spliced mRNAs in KO compared to WT testes. While these genes may be required for different aspects of spermatogenesis, the authors manuscript would be significantly strengthened if they can provide any evidence that the splicing changes induced by hnRNP GT loss negatively impact gene functions.

In this regard, the authors claim that "…insertion of this Kdm4d cryptic exon disables Kdm4d gene expression", citing the reduced read density on Kdm4d exon 2 shown in Figure 4A. I was not convinced that the splicing of the cryptic exon 'disables… gene expression' for the following reasons. First, clearly there are reads mapping to the exon2 indicating that full length transcripts are being produced. Second, because of the different scales used in the top and bottom RPKM tracks, it is difficult to state unequivocally that there is an appreciable difference in read density on exon 2. Third, Kdm4d should be in the list of 45 genes with reduced RNA levels (it is not in Figure 3—source data 1). Fourth, there is no western blot data showing that Kdm4d levels are reduced in KO testes. As it stands, the RT-PCR analysis of Kdm4d (primers in exon 1 and 2) is not adequate given the argument that polyadenylation at the cryptic exon primarily leads to truncation of transcripts. The authors should perform qRT-PCR to confirm decreased levels of the conventional transcript and ideally a western blot to show reduced protein expression.

The second example that is presented is alternative splicing of 3' terminal exons in Lrrcc1. The authors note that Lrrcc1 "is essential for mouse fertility (Figure 4D" but do not reference the publication. Do Lrrcc1 mutants have the same meiotic defect as hnRNP GT KO mice? In Figure 5, the authors describe changes in mRNA isoforms from the Meioc gene caused by activation of cryptic 3' and 5' splice sites. The authors spend a considerable amount of time carefully characterizing the splicing changes in Meioc mRNAs, but then ultimately indicate that "Consistent with the RNAseq data, expression levels of this full length Meioc protein did not change in the knockout (Figure 5D)." So again, I am left questioning whether the splicing in Meioc in hnRNP GT KO are functionally relevant and have any contribution to the hnRNP GT phenotype.

In light of these concerns it would greatly strengthen the authors' manuscript if they can provide data that more strongly support roles for the aforementioned (or other splicing changes) in the phenotypes observed in their hnRNP GT ko mice.

4) Whereas the RNAseq libraries are described in detail (ages, number of replicates, etc), there is scant information on the CLIP data. The authors are requested to add information on the age of samples, how many replicate tissues were used, how was the data processed and analyzed, how many genes have binding. It should also be mentioned how high-confidence binding sites were discriminated from experimental noise. Also, while AA-dinucleotides may be present in the top 10 pentamers, the quality of the CLIP information is unclear.

5) RNA binding maps representing different classes of splicing event would enhance the presentation of the CLIP analysis. For example, it would be interesting to see how hnRNP GT binding correlates with regulatory outcome from the RNA-seq analysis both for cryptic splice sites and for canonical types of alternative splicing like exon skipping events. Are similar features observed regarding mode of binding and regulation?

6) In Figure 6, the authors describe analysis of a minigene splicing reporter bearing the Kdm4d cryptic exon. A schematic of the minigene should be included in the Figure. The data shows that the cryptic exon is spliced in ~40% of reporter transcripts expressed in HEK293, and inclusion of the exon decreases significantly when hnRNP GT is expressed. A direct role for hnRNP GT in repressing the cryptic exon should be demonstrated by showing that mutation of AA dinucleotides abolishes hnRNP GT's ability to repress splicing (or bind the transcript).

Reviewer #1:

hnRNP GT is a mammalian autosomal paralog of RBMX and RBMY that is expressed during spermatogenesis when RBMX and RBMY are switched off. Its importance in spermatogenesis is indicated by mutations associated with male infertility.

In this manuscript David Elliott and colleagues use a mouse hnRNPGT conditional knockout model. As anticipated a very specific spermatogeneis phenotype was observed within exactly the time-window when hnRNPGT, but not RBMX or RBMY, is expressed. After careful and systematic phenotyping, testis mRNA-Seq was carried out at timepoint when the primary defect caused by hnRNPGT knockout could be observed (at later timepoints, the different cell-type compositions resulting from the initial phenotype would have dominated the signal). This showed that hnRNPGT deletion led to a large number of alternative splicing changes, with a strong enrichment for non-annotated (ie cryptic) events. The majority of cases would lead to down-regulated protein expression/activity, suggesting a compound phenotype arising from reduced expression of numerous proteins that are important during meiosis.

Overall, this is a carefully conducted investigation, clearly presented, which should be of interest to the readership of *eLife*.

Subsection “hnRNP GT protein controls splicing patterns during meiosis”.. The manuscript states that 23 of 27 tested splicing events were validated, citing Source Data 3. But this file has no data – only has a list of primers used for RT-PCR. It would be appropriate to include the results of the RT-PCR – as PSI (mean {plus minus} sd) for WT and KO conditions, and to define how an event was defined as validated. Did validation only require a statistically significant change in the same direction as for RNA-Seq? Ideally show the RT-PCR values alongside the MAJIQ values for PSI.

The RT-PCR examples shown in Figure 4B, Figure 5B and Figure 5—figure supplement 2A,B, all look very convincing and do not need quantitation to be convincing. Nevertheless, it would be good to show the PSI numbers to allow direct comparison with the MAJIQ output.

Reviewer #2:

In this report, Ehrmann et al., examine the functions of the RNA binding protein hnRNP GT in spermatogenesis. This protein is highly conserved across ~65 million years of evolution, is expressed during the meiotic and post-meiotic steps of spermatogenesis, and its’ in vivo roles have remained poorly understood.

The authors demonstrate that deletion of the hnRNP GT results in a major arrest of germ cell development during prophase of meiosis, with very few cells surviving to the post-meiotic (spermatid) stage. More detailed analyses indicate that most germ cells lacking hnRNP GT arrest at the diplotene stage of meiotic prophase. Experiments characterizing the phenotype of hnRNP GT knockout mice (KO) are well executed and interpreted reasonably.

To understand what gene expression events are altered in the absence of hnRNP GT, the authors perform RNA-Seq analysis on biologic triplicate testes from wild type (WT) and knockout (KO) mice collected at postnatal day 18. In contrast to the phenotypic analyses, I was disappointed with the RNA analyses described in the paper, as well as the authors' interpretation of the data.

1) Many genes are identified with differences in overall transcript abundance and/or changes in alternatively spliced isoforms. The authors largely ignore the 45 genes identified with differences in RNA level between WT and KO testis. There is no rationale presented for doing so. There are no descriptions of what the encoded proteins do, and how changes in their levels (caused by hnRNP GT KO) might contribute to the meiotic defects observed. Instead, the authors end this section rather abruptly and move on to the analysis of splicing changes. Why are these gene expression changes dismissed? This was somewhat surprising given that the authors mention (in the introduction) a role for RBMX in controlling transcription patterns.

2) Similarly, the analysis of alternative splicing changes are not explored in sufficient depth to provide new insight into why KO germ cells arrest in meiosis. Instead, the authors focus on the presentation of select examples of genes previously shown to be important for spermatogenesis and that also express alternatively spliced mRNAs in KO compared to WT testes. While these genes may be required for different aspects of spermatogenesis, there is no data presented to show that the splicing changes induced by hnRNP GT loss negatively impact gene functions.

The first of the examples presented is Kdm4d, in which loss of hnRNP GT results in splicing of a cryptic exon. The authors claim that "…insertion of this Kdm4d cryptic exon disables Kdm4d gene expression", citing the reduced read density on Kdm4d exon 2 shown in Figure 4A. I was not convinced that the splicing of the cryptic exon 'disables… gene expression' for the following reasons. First, clearly there are reads mapping to the exon2 indicating that full length transcripts are being produced. Second, because of the different scales used in the top and bottom RPKM tracks, it is difficult to state unequivocally that there is an appreciable difference in read density on exon 2. Third, Kdm4d should be in the list of 45 genes with reduced RNA levels (it is not in Figure 3*—*source data 1). Fourth, there is no western blot data showing that Kdm4d levels are reduced in KO testes.

The second example that is presented is alternative splicing of 3' terminal exons in Lrrcc1. The authors note that Lrrcc1 "is essential for mouse fertility (Figure 4D" but do not reference the publication. Do Lrrcc1 mutants have the same meiotic defect as hnRNP GT KO mice? The magnitude of the splicing change in Lrrcc1 is modest, as most AS changes tend to be. I'm not criticizing the authors for this, but unless the authors can show that this splicing change impacts Lrrcc1 function or germ cell progression through prophase, I am left shrugging my shoulders. The same can be said for references to Slc39a8.

In Figure 5, the authors describe changes in mRNA isoforms from the Meioc gene caused by activation of cryptic 3' and 5' splice sites. The authors spend a considerable amount of time carefully characterizing the splicing changes in Meioc mRNAs, but then ultimately indicate that "Consistent with the RNAseq data, expression levels of this full length Meioc protein did not change in the knockout (Figure 5D)." So again, I am left questioning whether the splicing in Meioc in hnRNP GT KO are functionally relevant and have any contribution to the hnRNP GT phenotype.

To conclude point 2, unless the authors can show that the highlighted splicing examples shown are biologically important and contribute to the hnRNP GT phenotype, the significance is doubted, particularly for Meioc, whose protein levels are reportedly unaltered.

3) Whereas the RNAseq libraries are described in detail (ages, number of replicates, etc), there is scant information on the CLIP data. What age was this performed on, how many replicate tissues, how was the data processed and analyzed, how many genes have binding? There is no mention of how high-confidence binding sites are discriminated from experimental noise. Thus, while AA-dinucleotides may be present in the top 10 pentamers, the quality of the CLIP information is unclear.

4) In Figure 6, the authors describe analysis of a minigene splicing reporter bearing the Kdm4d cryptic exon. A schematic of the minigene should be included in the Figure. The data shows that the cryptic exon is spliced in ~40% of reporter transcripts expressed in HEK293, and inclusion of the exon decreases significantly when hnRNP GT is expressed. A direct role for hnRNP GT in repressing the cryptic exon should be demonstrated by showing that mutation of AA dinucleotides abolishes hnRNP GT's ability to repress splicing (or bind the transcript).

5) The authors claim that hnRNP GT has a major role in protecting the meiotic transcriptome from aberrant selection of cryptic splice sites that are normally ignored by the spliceosome. While this may be a role, it's not clear that it is a major role. My reservation comes from the way in which these splicing events were selected. The authors state that "60% of the 87 most clearly visualized classical splicing events controlled by hnRNP GT involved the altered selection of cryptic splice sites." What is a 'clearly visualized' splicing event? 60% of 87 is 52, meaning that 52/237 splicing events (or 22% total) are cryptic splicing events.

6) "the poor conservation of splicing patterns between species has previously suggested alternative splicing might not have a major role in controlling fundamental aspects of germ cell biology (Kan et al., 2005)." The referenced paper compares human and mouse splicing across tissues. Besides meiosis, there are considerable differences in the mitotic and post-meiotic stages of spermatogenesis between human and mouse. Furthermore, several labs, including the corresponding author's, have provided strong evidence indicating important roles for regulated alternative splicing events in germ cell development. The authors should reference these studies.

In summary, Ehrmann and colleagues demonstrate that hnRNP GT is required for sperm formation, and hnRNP GT loss is associated with changes in RNA levels and alternatively spliced isoforms. However, the data is not explored in sufficient depth to link these two main observations, nor does it provide information on the mechanism(s) by which hnRNP GT functions on different RNAs. As a result, the data does not break new ground and the manuscript is not likely to appeal to the broad readership of *eLife*.

Reviewer #3:

In this manuscript, Ehrmann and colleagues characterize the germ cell-specific RNA binding protein hnRNP GT. The authors generated a conditional knockout mouse targeting hnRNP GT, and demonstrate significant defects in testis histology and spermatogenesis in mutant animals. Specifically, the authors demonstrate that defects in knockout animals likely stem from problems during a specific stage of meiosis. Given the observed phenotypes, transcriptome-wide analysis is performed looking both at steady-state mRNA levels as well as splicing patterns. The authors find both transcript-level and splicing-level differences between wild type and mutants, but their data suggest more prevalent defects in the latter layer of gene regulation. In particular, in addition to aberrant regulation of annotated alternative exons, the authors identify a more significant proportion of cryptic splice sites being utilized in mutant animals, and the use of these splice sites are predicted to have deleterious effects on encoded protein products. Additional experiments include HITS-CLIP assays that determine the sequence binding preference of hnRNP GT (motifs with AA dinucleotides) and information about direct targets of this RNA binding protein. Finally, the authors show in cell lines that over-expression of hnRNP GT has the ability to suppress usage of a cryptic splice site, even in the context of co-expression of the activating SR protein Tra2β.

Overall, I found the study very interesting and well executed, with very clean results and convincing phenotypes. It remains an important goal for continued characterization of RNA binding proteins in tissue- and context-specific alternative splicing, and relative to other tissues like the nervous system and muscle cells, there is comparatively little known about germ cell regulators of splicing.

I have only a few points that would be nice to see addressed by the authors:

1) The authors indicate that hnRNP GT, RBMX and RBMY are paralogs. It would be nice to see a sequence alignment of all three proteins from various species to better visualize this. For example, what type of conservation is observed over relevant domains in the three RNA binding proteins?

2) RNA maps would be nice to include as part of the CLIP analysis, perhaps separated by class of splicing event. For example, it would be interesting to see how hnRNP GT binding correlates with regulatory outcome from the RNA-seq analysis both for cryptic splice sites and for canonical types of alternative splicing like exon skipping events. Are similar features observed regarding mode of binding and regulation?

3) Should all of the 'cryptic' splice sites really be annotated as 'cryptic'? This is perhaps more about semantics, but it sounds like the authors define cryptic splice sites as anything that is not annotated in the mm10 gene annotations. However, most RNA-seq data (even from wild type animals) will identify novel junctions often not documented in existing annotations on databases (especially tissue-specific isoforms which are often functional).

Perhaps this is just a matter of semantics, but in my opinion, cryptic splicing would involve novel splice sites that are never detected in wild type animals, as opposed to minor isoforms that are still found in wild type animals.

Some clarification from the authors would be nice regarding this point regardless of what they decide to call these variants. Specifically, are the majority of the cryptic events they detect in mutants completely absent in wild type animals, or are they detectable in wild type, just at low abundance?

---

## [Author Response]

[Editors’ notes: the authors’ response after being formally invited to submit a revised submission follows.]

In this manuscript, Ehrmann and colleagues characterize the germ cell-specific RNA binding protein hnRNP GT. The authors generated a conditional knockout mouse targeting hnRNP GT, and demonstrate significant defects in testis histology and spermatogenesis in mutant animals. Specifically, the authors demonstrate that defects in knockout animals likely stem from problems during a specific stage of meiosis. Given the observed phenotypes, transcriptome-wide analysis is performed looking both at steady-state mRNA levels as well as splicing patterns. The authors find both transcript-level and splicing-level differences between wild type and mutants, but their data suggest more prevalent defects in the latter layer of gene regulation. In particular, in addition to aberrant regulation of annotated alternative exons, the authors identify a more significant proportion of 'cryptic'/minor splice sites being utilized in mutant animals, and the use of these splice sites are predicted to have deleterious effects on encoded protein products. Additional experiments include HITS-CLIP assays that determine the sequence binding preference of hnRNP GT (motifs with AA dinucleotides) and information about direct targets of this RNA binding protein. Finally, the authors show in cell lines that over-expression of hnRNP GT has the ability to suppress usage of a cryptic splice site, even in the context of co-expression of the activating SR protein Tra2β.Overall, this is a carefully conducted investigation that should be of considerable interest to the readership of eLife after suitable revision. It is requested that authors provide a proposed plan for revisions that can be performed within approximately two months, based on the comments below. In particular, it was felt that the paper would be significantly strengthened by the inclusion of additional data supporting functional roles for the isoforms highlighted by the authors, and in this regard their plan should pay special attention to point 3 below. It was also felt that the section of the paper relating to functional consequences of isoform switches could be better focused and the authors should make an effort to revise this section accordingly.Essential revisions:1) Subsection “hnRNP GT protein controls splicing patterns during meiosis”. The manuscript states that 23 of 27 tested splicing events were validated, citing Source Data 3. But this file has no data – only has a list of primers used for RT-PCR. It would be appropriate to include the results of the RT-PCR – as PSI (mean {plus minus} sd) for WT and KO conditions, and to define how an event was defined as validated. Did validation only require a statistically significant change in the same direction as for RNA-Seq? Ideally show the RT-PCR values alongside the MAJIQ values for PSI.The RT-PCR examples shown in Figure 4B, Figure 5B and Figure 5—figure supplement 2A,B, look convincing and do not need quantitation to be convincing. Nevertheless, it would be good to show the PSI numbers to allow direct comparison with the MAJIQ output.

Good suggestion, we added this data to the paper as an extra column to Figure 3*—*source data 4, so it is right next to the primers.

2) Many genes are identified with differences in overall transcript abundance and/or changes in alternatively spliced isoforms. The authors largely ignore the 45 genes identified with differences in RNA level between WT and KO testis. The authors should comment on these genes and provide information on what the encoded proteins do, and how changes in their levels (caused by hnRNP GT KO) might contribute to the meiotic defects observed.

We now add this extra information and discuss the transcriptional changes in the text as requested. There are some interesting genes in the list of transcriptional changes. These include Cul4a, constitutive knockout of which causes a meiotic phenotype, and Fsip2 which encodes a component of the sperm midpiece. We also added qPCR analysis here to confirm gene expression changes of Fsip2 (new Figure 3*—*figure supplement 1). We agree with the reviewer that addition of this extra material makes our analysis much more complete. We also discuss how constitutive knockouts of genes needed in late meiosis may arrest at earlier stages in germ cell development if they are also needed earlier. This means that later stages will be invisible in terms of the knockout phenotype (please see also below).

3) The analysis of alternative splicing changes is not explored in sufficient depth to provide insight into why KO germ cells arrest in meiosis. The authors focus on select examples of genes previously shown to be important for spermatogenesis and that also express alternatively spliced mRNAs in KO compared to WT testes. While these genes may be required for different aspects of spermatogenesis, the authors manuscript would be significantly strengthened if they can provide any evidence that the splicing changes induced by hnRNP GT loss negatively impact gene functions.In this regard, the authors claim that "…insertion of this Kdm4d cryptic exon disables Kdm4d gene expression", citing the reduced read density on Kdm4d exon 2 shown in Figure 4A. I was not convinced that the splicing of the cryptic exon 'disables… gene expression' for the following reasons. First, clearly there are reads mapping to the exon2 indicating that full length transcripts are being produced. Second, because of the different scales used in the top and bottom RPKM tracks, it is difficult to state unequivocally that there is an appreciable difference in read density on exon 2. Third, Kdm4d should be in the list of 45 genes with reduced RNA levels (it is not in Figure 3—source data 1). Fourth, there is no western blot data showing that Kdm4d levels are reduced in KO testes. As it stands, the RT-PCR analysis of Kdm4d (primers in exon 1 and 2) is not adequate given the argument that polyadenylation at the cryptic exon primarily leads to truncation of transcripts. The authors should perform qRT-PCR to confirm decreased levels of the conventional transcript and ideally a western blot to show reduced protein expression.

We agree with the reviewer that from just the UCSC screenshot originally shown (Figure 4 in the manuscript) that this Kdm4d cryptic exon does not stand out as a terminal exon. We also carried out more detailed analysis of junction reads. This supports our interpretations, and we now incorporate this extra data into

the manuscript (in the new Figure 4—figure supplement 2). In the Sashimi plot

(Figure 4—figure supplement 2A), all wild type junction reads (71) join directly exon 1 to exon 2 (wild type sample shown in red, knockout sample shown in blue, and the Kdm4d gene reads from right to left). In contrast, the knockout sample shows a more complex pattern across the first intron. Importantly, almost all the reads terminate at the cryptic exon. There are 140 reads joining exon 1 to the cryptic exon. Only 10 junction reads join the cryptic exon to exon 2. Although we only show data for one knockout and one wild type mouse on the Sashimi plot, we saw the same patterns for exon junction reads for each of the knockout mice and wild type mice. The most straightforward conclusion to this exon junction read pattern is that the Kdm4d cryptic exon operates as a terminal exon.

In our expression analysis shown in Figure 3A, Kdm4d shows a log2 fold change in expression of -0.96 with a p value of 0.00048, but the padj value is 0.129 so this did not qualify for significance. We now provide the full table of expression levels between wild type and knockout for all genes as source data (see new Figure 3*—*source data 1). The lack of statistical significance is because of biological variability between mice. We also tested expression of Kdm4d exon 2 using qPCR. A decrease in Kdm4d exon 2 expression was detected across all knockout compared to wild type mice (new Figure 4—figure supplement 2B). Similar to the DESeq analysis this did not reach statistical significance, but it illustrates that the trend was the same in all 3 knockout mice.

Published antibodies raised against mouse Meioc enabled us to detect a shorter

protein isoform only in the knockout and not in the wild type or heterozygote (see also below). However, there is in general lack of validated antibodies against the mouse proteins that we would like to investigate. There is an Abcam antibody reported to react with mouse Kdm4d, but the single paper that used it does not show a Western of Kdm4d protein. We proposed in our action plan to obtain a new commercial (Proteintech) antibody specific for BRCA2 that is predicted to recognize mouse BRCA2 protein (384KDa). We tested this antibody by Western, but it mainly detected proteins of 250KDa and 70KDa even in wild type testis - much smaller than that corresponding to the expected size of full length BRCA2 protein. These data raise questions about the specificity of this antibody in mice, so we have not included this data in our revised manuscript.

The second example that is presented is alternative splicing of 3' terminal exons in Lrrcc1. The authors note that Lrrcc1 "is essential for mouse fertility (Figure 4D" but do not reference the publication. Do Lrrcc1 mutants have the same meiotic defect as hnRNP GT KO mice?

In our updated manuscript we have made it clearer how we accessed the MGI and IMPC phenotypes, by referring to the Mouse Genome Database. For example, Lrrcc1 has a male infertile phenotype from the mouse knockout project (http://www.ensembl.org/Mus_musculus/Gene/Phenotype?db=core;g=ENSMUSG00 000027550;r=3:14533788-14572658). We cite the Smith el al., 2018. This paper is about the database which we used to get the information, not specifically about just Lrrcc1 itself. We also made it clearer in our revised text that (1) the testis phenotype characterisation available for gene knockouts available via MGI and IMKP are not as detailed as we have performed for Hnrnpgt, so it is difficult to side by side compare them; and that (2) because of its expression profile, Hnrnpgt is likely to be most important for control of its target genes well after meiotic prophase has started –even if these target genes are also expressed earlier in germ cell development. In contrast, the mouse knockout phenotypes reported at MGD are often constitutive knockouts, so will arrest germ cell development the first time that the target gene is needed –often earlier than late prophase. For example, the phenotype of the constitutive Meioc knockout mouse has focussed on meiotic entry, well before late meiotic prophase when the Hnrnpgt KO mouse has a phenotype. So while Meioc is needed early in meiosis, a later role in meiosis is currently been invisible within the Meioc knockout mouse. Hence the published constitutive knockout does not tell us how impairing the function of Meioc during late pachytene would affect spermatogenesis. We also discuss the possibility that Rbmx and Rbmy could provide an equivalent function to Hnrnpgt that operates in spermatogonia and early spermatocytes, and that it is only when meiotic sex chromosome inactivation kicks in during pachytene that any critical non-redundant functions of Hnrnpgt will become apparent. We tested RBMX protein control of the Kdm4d cryptic exon in our revised Figure 6B.

In Figure 5, the authors describe changes in mRNA isoforms from the Meioc gene caused by activation of cryptic 3' and 5' splice sites. The authors spend a considerable amount of time carefully characterizing the splicing changes in Meioc mRNAs, but then ultimately indicate that "Consistent with the RNAseq data, expression levels of this full length Meioc protein did not change in the knockout (Figure 5D)." So again, I am left questioning whether the splicing in Meioc in hnRNP GT KO are functionally relevant and have any contribution to the hnRNP GT phenotype.In light of these concerns it would greatly strengthen the authors' manuscript if they can provide data that more strongly support roles for the aforementioned (or other splicing changes) in the phenotypes observed in their hnRNP GT ko mice.

The referee raises the important point of what is the importance of the

cryptic spliced isoforms if the normal mRNAs are also expressed in the testis. We

think this comes down to which cells express particular isoforms, and have carried out more experimental analysis to address this. Since hnRNP GT is only expressed in meiotic prophase, and its knockout has a late prophase arrest, we considered whether the correctly spliced Meioc exon 5 detected in total testis RNA could originate from the many cell types that exist before prophase. If this was the case, then the normal splicing signal from earlier germ cell types would make a significant contribution to swamping out the signal detected from aberrant splicing patterns in later germ cell types (including cells in diplotene).

We tested this by monitoring whether levels of cryptic splicing patterns become

stronger as the cell types that depend more exclusively on hnRNP GT expression

appear. This time-course data is shown in the new Figure 5E, and shows that

cryptic splicing of Meioc exon 5 does get progressively worse during meiotic

prophase. The normal splice isoform of Meioc was strongly expressed in the 10-

13dpp testis, with the shorter isoforms virtually invisible. 13dpp testis contain type B spermatogonia, preleptotene spermatocytes, zygotene spermatocytes and cells entering pachytene (Bellve et al., 1977). Just 3 days later, cryptic splicing patterns are easily detectable within 16dpp testes (which contain cells within pachytene, as well as the type B spermatogonia, preleptotene spermatocytes, zygotene spermatocytes found in 13dpp testis). These data support a model where cryptic Meioc splicing occurs within the specific cell types that rely on hnRNP GT for RNA processing, and that the normal splice isoform signal in whole testis RNA must be coming at least in part from earlier germ cell types before meiotic prophase.

Likewise, the shorter Meioc protein detected by Western could be specifically from cells in late prophase, and full length from earlier germ cell types (Figure 5D).

We also performed similar time-course experiments to analyse cryptic splicing of Brca2. This showed the same picture, that the Brca2 cryptic splice isoform comes from cells in meiotic prophase, and that there is a strong signal from normally spliced Brca2 from earlier germ cell types (new Figure 5*—*figure supplement 2).

We have also added additional discussion to our manuscript. Since we observe

splicing defects in 186 genes, the phenotype we observe may likely be due to the

additive loss of several or many genes.

4) Whereas the RNAseq libraries are described in detail (ages, number of replicates, etc), there is scant information on the CLIP data. The authors are requested to add information on the age of samples, how many replicate tissues were used, how was the data processed and analyzed, how many genes have binding. It should also be mentioned how high-confidence binding sites were discriminated from experimental noise. Also, while AA-dinucleotides may be present in the top 10 pentamers, the quality of the CLIP information is unclear.

The CLIP analysis was carried out using one adult mouse, and was processed via the standard pipeline by Tomaz Curk using the icount software (http://icount.fri.uni-lj.si). We have now added in this extra information to the Methods section of our manuscript, and referenced the appropriate publication for icount.

5) RNA binding maps representing different classes of splicing event would enhance the presentation of the CLIP analysis. For example, it would be interesting to see how hnRNP GT binding correlates with regulatory outcome from the RNA-seq analysis both for cryptic splice sites and for canonical types of alternative splicing like exon skipping events. Are similar features observed regarding mode of binding and regulation?

We added this data as new Figure 3*—*figure supplement 4. We carried out this analysis on our current HITS-CLIP data, but also added in an analysis using the predicted binding site from the HITSCLIP data as well.

6) In Figure 6, the authors describe analysis of a minigene splicing reporter bearing the Kdm4d cryptic exon. A schematic of the minigene should be included in the Figure. The data shows that the cryptic exon is spliced in ~40% of reporter transcripts expressed in HEK293, and inclusion of the exon decreases significantly when hnRNP GT is expressed. A direct role for hnRNP GT in repressing the cryptic exon should be demonstrated by showing that mutation of AA dinucleotides abolishes hnRNP GT's ability to repress splicing (or bind the transcript). We propose to test this using an RRM negative hnRNPGT.

We added the schematic (new Figure 6A). In fact, we don’t detect CLIP tags near the Kdm4d cryptic exon, although we do see them elsewhere on the gene (Figure 4A). Although lack of tags does not definitively exclude hnRNP GT binding to this region of the Kdm4d gene, it does suggest that hnRNP GT control of this cryptic exon could be largely through protein-protein interactions with another

splicing regulator such as Tra2β, rather than direct RNA-protein binding. We tested this idea by co-transfection experiments with full length hnRNP GT, and with an hnRNP GT variant from which the RNP1 motif in the RRM (RNA recognition motif) had been deleted (updated Figure 6B). Since both co-transfections blocked splicing inclusion of this cryptic exon this experiment supports the effect of hnRNP GT being largely not mediated by direct protein-RNA interactions by the RRM. Interestingly, although the ΔRNP1 version of hnRNP GT is able to repress splicing inclusion of the Kdm4d exon, it did not work quite as efficiently as the full length version of the protein. This leaves open some role for the RRM in regulating this Kdm4d exon.

We have also added in new data showing that RBMX is able to repress splicing of

the Kdm4d cryptic exon (updated Figure 6B). This supports the idea that RBMX

protein may fulfil the function of hnRNP GT prior to meiotic prophase, and so

contribute to the normal splicing patterns observed from testes before 14dpp.